# Homologous chromosomes are stably conjoined for *Drosophila* male meiosis I by SUM, a multimerized protein assembly with modules for DNA-binding and for separase-mediated dissociation co-opted from cohesin

Zeynep Kabakci[1], Heidi E. Reichle[2], Bianca Lemke[2], Dorota Rousova[2], Samir Gupta[1], Joe Weber[1], Alexander Schleiffer[3], John R. Weir[2]*, Christian F. Lehner[1]*

**1** Department of Molecular Life Science (DMLS), University of Zurich, Zurich, Switzerland, **2** Friedrich Miescher Laboratory of the Max Planck Society, Tübingen, Germany, **3** Research Institute of Molecular Pathology (IMP), Vienna BioCenter, Vienna, Austria

* john.weir@tuebingen.mpg.de (JRW); christian.lehner@imls.uzh.ch (CFL)

## Abstract

For meiosis I, homologous chromosomes must be paired into bivalents. Maintenance of homolog conjunction in bivalents until anaphase I depends on crossovers in canonical meiosis. However, instead of crossovers, an alternative system achieves homolog conjunction during the achiasmate male meiosis of *Drosophila melanogaster*. The proteins SNM, UNO and MNM are likely constituents of a physical linkage that conjoins homologs in *D. melanogaster* spermatocytes. Here, we report that SNM binds tightly to the C-terminal region of UNO. This interaction is homologous to that of the cohesin subunits stromalin/Scc3/STAG and α-kleisin, as revealed by sequence similarities, structure modeling and cross-link mass spectrometry. Importantly, purified SU_C, the heterodimeric complex of SNM and the C-terminal region of UNO, displayed DNA-binding *in vitro*. DNA-binding was severely impaired by mutational elimination of positively charged residues from the C-terminal helix of UNO. Phenotypic analyses in flies fully confirmed the physiological relevance of this basic helix for chromosome-binding and homolog conjunction during male meiosis. Beyond DNA, SU_C also bound MNM, one of many isoforms expressed from the complex *mod(mdg4)* locus. This binding of MNM to SU_C was mediated by the MNM-specific C-terminal region, while the purified N-terminal part common to all Mod(mdg4) isoforms multimerized into hexamers *in vitro*. Similarly, the UNO N-terminal domain formed tetramers *in vitro*. Thus, we suggest that multimerization confers to SUM, the assemblies composed of SNM, UNO and MNM, the capacity to conjoin homologous chromosomes stably by the resultant multivalent DNA-binding. Moreover, to permit homolog separation during anaphase I, SUM is dissociated by separase, since UNO, the α-kleisin-related protein, includes a separase cleavage site. In support of this proposal, we demonstrate that UNO cleavage by tobacco etch virus protease is sufficient to release homolog conjunction *in vivo* after mutational exchange of the separase cleavage site with that of the bio-orthogonal protease.

**Data Availability Statement:** All relevant data are within the manuscript and its Supporting information files.

**Funding:** The research was supported by funds obtained from the Swiss National Science Foundation (www.snf.ch), grant number 31003A_179433 (CFL), from the Max-Planck Society (JRW) and from the German Research Foundation (www.dfg.de), grant number WE 6513/2-1 (JRW). The funders had no role in study design, data collection and analysis, decision to publish, or preparation of the manuscript.

**Competing interests:** The authors have declared that no competing interests exist.

## Author summary

A major purpose of meiosis is the reduction of genome ploidy from diploid to haploid. Therefore, homologous chromosomes are paired into bivalents before the first meiotic division. Conjunction of homologs into bivalents usually results from crossovers (generated by meiotic recombination) in combination with sister chromatid cohesion by cohesin complexes that preclude crossover terminalization. Cohesin is thought to form a proteinaceous ring around sister chromatids. After bivalent biorientation in the spindle, a particular cohesin subunit is cleaved by the endoprotease separase, allowing homolog separation during the first meiotic division. Surprisingly, homolog conjunction during meiosis in males of the fly *Drosophila melanogaster* does not involve crossover formation. Instead, an alternative homolog conjunction system is used. Here we report how the proteins of this alternative system form assemblies that bind DNA. Conjunction of homologous chromosomes in *D. melanogaster* males depends on these protein assemblies, functioning apparently akin to conventional glue rather than as topological embrace like cohesin. However, after bivalent biorientation, separase can be used nevertheless for efficient dissolution of the alternative homolog glue in *D. melanogaster* spermatocytes, as some of the alternative conjunction proteins have evolved from cohesin subunits.

## Introduction

Regular chromosome transmission during mitotic and meiotic divisions depends on sister chromatid cohesion. The corresponding ties that keep sister chromatids paired are formed already during S phase, concomitant with chromosome replication. They consist primarily of cohesin, a protein complex based on three core subunits, SMC1, SMC3 and an α-kleisin [1]. The latter provides binding sites for additional cohesin subunits that are known as HAWK proteins (HEAT repeat proteins Associated With Kleisins) [1,2]. One of the HAWKs, a member of the stromalin protein family, which includes Scc3 of budding yeast, *Drosophila* SA-1 and the human proteins STAG1-3 (also designated as SA1-3), is permanently bound. Moreover, Pds5 or Scc2/Nipped-B/NIPBL proteins are recruited interchangeably by another binding site on α-kleisin.

Cohesin might mediate sister chromatid cohesion in a topological manner, by forming a proteinaceous ring around the sister chromatid pair [1]. Thereby, this topological embrace also occasions an effective way of releasing sister chromatid cohesion by proteolytic opening of the ring. Evidently, after biorientation of all chromosomes within the spindle, cohesion between sister chromatids needs to be severed during mitosis, as well as during the second meiotic division (M II), to permit the segregation of sister centromeres to opposite spindle poles. The final release of sister chromatid cohesion is known to be promoted by the endoprotease separase, which cleaves specifically the α-kleisin subunit of cohesin after separase activation at the metaphase-to-anaphase transition [3–6].

Sister chromatid cohesion is also essential for the regular segregation of homologous chromosomes during the first meiotic division (M I). In combination with crossovers (COs), cohesion of sister chromatids in chromosome arm regions on the telomeric side of COs precludes CO terminalization. Therefore, the combination of COs and distal sister chromatid cohesion maintains paired homologs as bivalent chromosomes during canonical meiosis [7–11]. After biorientation of all the bivalents in the M I spindle, homolog separation at the metaphase-to-anaphase transition is also induced by separase [9–11], similar as in mitosis and M II.

However, α-kleisin cleavage by separase during M I is spatially controlled [11,12]. While α-kleisin of cohesin on chromosome arms is cleaved during M I, it is protected from cleavage by separase within pericentromeric regions. The spatial control of cohesin elimination during M I depends on the expression of Rec8-type, meiosis-specific α-kleisins and their spatially differentiated phosphorylation, while the Scc1/Rad21 proteins function as α-kleisins during mitosis.

Strikingly, male meiosis in higher dipteran species including *Drosophila melanogaster* is achiasmate [13,14]. Neither the formation of a synaptonemal complex, which mediates synapsis of homologous chromosomes all along their length during canonical meiosis, nor meiotic recombination and CO formation occur. Nevertheless, homologous chromosomes are also paired into bivalents before separation to opposite spindle poles during anaphase of M I. The initial pairing of homologous chromosomes in spermatocytes might be driven by the same mechanisms that mediate the wide-spread pairing of homologous chromosomes in somatic cell types of *D. melanogaster* [15,16]. However, somatic pairing appears to be disrupted by chromosome condensation at the onset of mitotic divisions [17,18]. In contrast, in spermatocytes, homolog pairing is maintained until the onset of anaphase I by a special conjunction system that functions as a replacement of COs [13]. Four genes have been identified as specifically required for this alternative homolog conjunction (AHC) [19–22]. Null mutations in these AHC genes result in premature separation of bivalents into univalents and random segregation of the univalents during M I in males, but not in females where meiosis is canonical. The AHC gene *teflon* (*tef*) is required predominantly for normal formation and segregation of autosomal bivalents [19]. In contrast, the three additional AHC genes, *stromalin in meiosis* (*snm*) (or also *SA-2*), *modifier of mdg in meiosis* (*mnm*) and *univalents only* (*uno*), are equally important for autosomal bivalents and the sex chromosome bivalent [20,22].

The molecular details of how the AHC proteins contribute to maintenance of chromosome conjunction in bivalents are still poorly understood. TEF appears to function in early spermatocytes, during establishment of AHC [21,23]. In late spermatocytes, TEF is no longer detectable [23]. Thus, it does not appear to be a component of the chromosomal glue that mediates AHC until anaphase I. TEF contains three zinc fingers [21]. It can bind to chromosomes and recruit MNM, which binds directly to TEF [23].

In contrast to TEF, MNM as well as the additional AHC proteins SNM and UNO are presumably part of the chromosome conjunction glue. SNM, UNO and MNM, abbreviated as SUM in the following, are co-localized on bivalents [20,22]. Their presence is particularly prominent on the sex chromosome bivalent. While the sex chromosomes of *D. melanogaster* do not share any extended euchromatic homology, they both contain rDNA repeats embedded in centromere-proximal heterochromatin, and these rDNA loci serve as pairing centers during male meiosis [24]. In particular, a 240 bp repeat normally located within the intergenic spacers of the rDNA repeats was demonstrated to be sufficient for chromosome conjunction [20,24–26]. The SUM proteins are associated with the 240 bp repeats in sub-nucleolar foci in mid to late spermatocytes [20]. During chromosome condensation at the onset of the first meiotic division (M I), the sub-nucleolar SUM foci coalesce into a single large dot at the XY chromosome (chr) pairing site [20,22]. On autosomal bivalents, the SUM proteins are present in much smaller dots that are not necessarily always detectable above background [20,22,23,27]. The number of SUM dots on autosomal bivalents appears to be restricted to around one or two per bivalent at the start of M I [23,27]. Their chromosomal localization is presumably distinct in each spermatocyte, as genetic analyses have revealed a wide-spread distribution of meiotic conjunction potential throughout the euchromatic regions rather than dedicated invariant autosomal conjunction sites [28,29]. Possibly, therefore, AHC is established during spermatocyte maturation in a rather randomly chosen restricted chromosomal region in case of autosomal bivalents, analogous to the placement of COs during canonical meiosis [27]. Strong

support for the notion that the SUM protein dots on the sex chromosome and autosomal biva-lents represent the physical conjunction between chromosomes is their precipitous, separase-dependent disappearance at the onset of anaphase I [20,22,30]. Mutational elimination of a separase cleavage site motif in UNO was shown to block the disappearance of the chromo-somal SUM protein dots as well as homolog separation during anaphase I [22].

For further clarification of the molecular mechanisms whereby the SUM proteins, which do not include well-known *bona fide* DNA-binding motifs, might achieve physical conjunction of chromosomes, we have initiated biochemical analyses of their interaction domains and their DNA-binding. Our findings reveal that co-option of the cohesin-related proteins SNM and UNO for AHC has conferred the ability to regulate chromosome conjunction by separase activity, as well as an ability to bind DNA. However, our observations indicate that SUM pro-teins do not conjoin chromosomes in a topological manner akin to cohesin. Rather, effective chromosome conjunction appears to be achieved by tight multivalent DNA-binding resulting from multimerization via both UNO and MNM.

## Results

### The C-terminal region of UNO that binds to SNM is derived from α-kleisins

UNO homologs can be detected exclusively within Diptera (suborder Brachycera). They have conserved N- and C-terminal domains separated by a putative disordered linker region (Fig 1A). UNO of *D. melanogaster* was identified originally because of co-purification with both SNM-EGFP and MNM-EGFP from testis extracts [22]. To assess whether SNM or MNM might bind directly to UNO, we performed co-immunoprecipitation experiments after tran-sient expression of tagged proteins in *Drosophila* S2R+ cells. In this cell line, endogenous UNO expression occurs at most at very low levels according to RNA-seq data, while SNM mRNA is not detectable [31,32]. Similarly, by immunoblotting, we failed to detect endogenously expressed UNO and SNM in S2R+ cells (S1 Fig). In case of MNM, which is but one of many distinct isoforms expressed from the complex *mod(mdg4)* locus [20,33], RNA-seq data is con-sistent with endogenous expression in S2R+ cells at low levels [31,32]. As antibodies specific for MNM are not available, the potential presence of endogenous MNM protein in S2R+ cells remains to be confirmed. However, immunoblotting with anti-M_CP, an antibody that reacts with the N-terminal common part (CP) present in all the different Mod(mdg4) isoforms [34], demonstrated that endogenous MNM, if present, is at a level far lower than that generated from transfected expression plasmids (S1 Fig).

After transfection of S2R+ cells with plasmids for co-expression of SNM-mCherry and UNO-EGFP, we observed efficient co-immunoprecipitation of these two proteins (Fig 1B). In contrast, co-immunoprecipitation of MNM-mCherry and UNO-EGFP could not be observed in an analogous experiment (Fig 1C).

According to our initial bioinformatic analyses with standard BLAST searches [22], the pre-dicted amino acid (aa) sequence of UNO failed to display significant similarities to proteins with known functions. However, because of the apparent direct binding of UNO to SNM, a member of the stromalin family of α-kleisin-binding proteins, and because UNO was previ-ously shown to include a functionally essential cleavage site cut by separase (Fig 1A) [22], which cleaves primarily α-kleisins, we focused specifically on detecting α-kleisin similarities within UNO. Interestingly, a region close to the C-terminus of UNO (aa 289–364) was found to be similar to an internal α-kleisins region (aa 320–394 of human Rad21) (Figs 1A and S2). This conserved internal α-kleisin region is known to bind stromalin/SA/STAG proteins [35,36]. We conclude that UNO includes a region that was co-opted from an α-kleisin. This

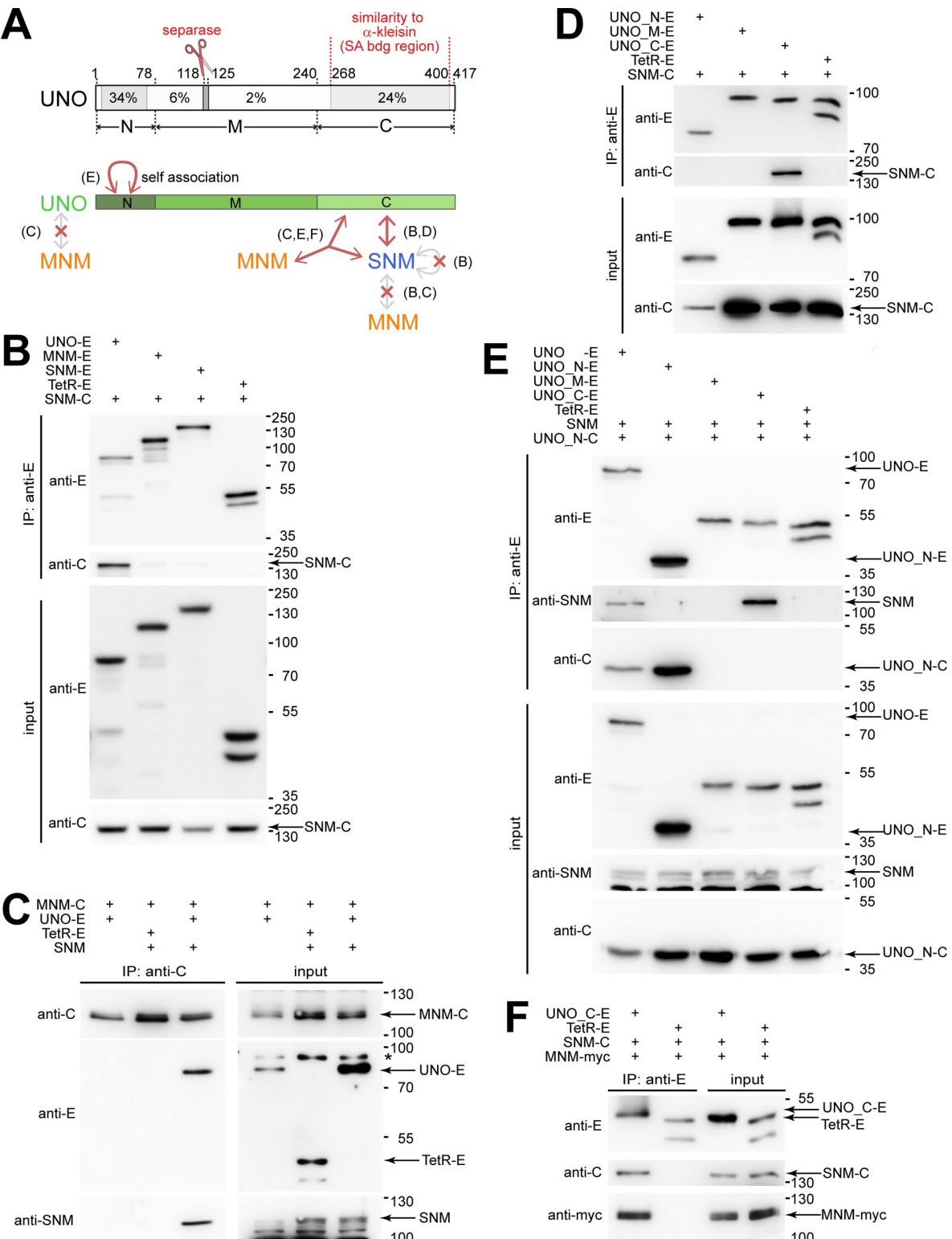

**Fig 1. SNM and UNO form a complex that binds MNM. (A)** Structure of UNO and interactions with AHC proteins. UNO includes two regions with stronger sequence conservation close to the N- and C-termini, as well as a highly conserved match to the separase cleavage-site consensus. Regional extent of amino acid identity (%) among drosophilid UNO orthologs is indicated with grey shading. Protein interactions of SNM, MNM and full-length UNO, as well as the UNO fragments UNO_N, UNO_M and UNO_C were analyzed by co-immunoprecipitation experiments after transient expression in S2R+ cells. The main findings are summarized schematically (with

reference to relevant figure panels in brackets). (**B-F**) Co-immunoprecipitation after transient expression of the indicated proteins was analyzed by immunoblotting. Protein tags were EGFP (E), mCherry (C) or a myc epitope (myc). Tetracycline repressor fused to a nuclear localization signal and EGFP (TetR-E) was used for control experiments. Presence or absence of proteins in the extracts used for immunoprecipitation (input) or in the samples immunoprecipitated with the indicated antibodies (IP) were analyzed with anti-EGFP (anti-E), anti-mCherry (anti-C), anti-myc (anti-myc) and anti-SNM. A non-specific band (*) recognized by anti-E and the positions of molecular weight markers are indicated on the right side, as well as the bands representing proteins of interest.

region of UNO comprises a separase cleavage site and the C-terminal stromalin-binding region. Co-immunoprecipitation experiments clearly confirmed that a C-terminal fragment of UNO (UNO_C, aa 241–417) mediates binding of UNO to SNM-mCherry (Fig 1A and 1D). The N-terminal region of UNO (UNO_N, aa 1–78) and the middle part (UNO_M, aa 79–240) did not bind to SNM (Fig 1A and 1D). We conclude that SNM and UNO appear to form a complex, which will be designated as SU complex in the following.

The co-immunoprecipitation experiments also indicated that SU complex formation was accompanied by mutual stabilization of the interacting proteins. After transient co-expression, the resulting levels of UNO and SNM were around sixfold higher compared to those detected after individual expression (S3 Fig). Analyses with UNO subregions indicated that UNO_C stability was highly dependent on the presence of SNM, while UNO_N and UNO_M appeared to be more stable and not dependent on SNM (S3 Fig).

## The N-terminal region of UNO mediates self-association

Other characteristic hallmarks of α-kleisins, the N- and C-terminal domains, which bind to the cohesin core subunits SMC3 and SMC1, respectively, are absent in UNO. UNO does not extend C-terminally beyond the SNM-binding region. At the N-terminus, UNO has a domain that is distinct from those present in α-kleisins. The N-terminal domain of UNO (UNO_N, aa 1–78) is predicted to form β-strands, while the N-terminal SMC3-binding region of α-kleisins is α-helical. Moreover, a region with high sequence similarity to UNO_N is present in a poorly characterized *Drosophila* protein (CG32117) that has no similarity to α-kleisins [22].

Interestingly, our co-immunoprecipitation experiments demonstrated that (UNO_N) mediates self-association. After co-expression of UNO_N-EGFP and UNO_N-mCherry, we observed their efficient co-immunoprecipitation (Fig 1A and 1E). UNO_N-mCherry was also co-immunoprecipitated by full-length UNO-EGFP along with SNM (Fig 1E). In contrast, only SNM but not UNO_N-mCherry was co-immunoprecipitated by UNO_C-EGFP (Fig 1E). These results suggest that the distinctive N-terminal domain of UNO (UNO_N) mediates self-association, even when SNM is bound to UNO. SNM does not appear to have the ability to self-associate, in contrast to UNO. We did not detect co-immunoprecipitation of SNM-EGFP and SNM-mCherry (Fig 1A and 1B).

## A complex of SNM and UNO binds MNM

The lack of co-immunoprecipitation of MNM-mCherry and UNO-EGFP (Fig 1C) suggested that the co-purification of MNM-EGFP and UNO from *Drosophila* testis extracts [22] was not the result of a direct interaction between these two proteins. Thus, we considered the possibility that SNM might function as a bridging factor, by binding not only to UNO, but perhaps also to MNM. However, SNM-mCherry binding to MNM-EGFP could not be detected in co-immunoprecipitation experiments (Fig 1A and 1B). Similarly, untagged SNM was not co-immunoprecipitated by MNM-mCherry (Fig 1C). However, when the three proteins SNM, UNO-EGFP and MNM-mCherry were co-expressed, we clearly detected their co-immunoprecipitation (Fig 1C). Moreover, we also observed efficient co-immunoprecipitation of the C-

terminal domain of UNO (UNO_C-EGFP) with SNM-mCherry and MNM-myc (Fig 1F). In conclusion, the SU complex formed by SNM and UNO can bind MNM, while individually SNM and UNO do not appear to have this ability (Fig 1A). The complex formed by the three proteins SNM, UNO and MNM will be designated as SUM complex in the following.

## MNM self-association permits binding of TEF to the SUM complex

MNM is only one of more than thirty distinct protein isoforms expressed by *mod(mdg4)* [20,34]. Mod(mdg4) products, including MNM, share an N-terminal common part (CP) followed by an isoform-specific C-terminal region (Fig 2A). To address whether SU binds only MNM or also other Mod(mdg4) proteins, we performed additional co-immunoprecipitation experiments. The Mod(mdg4) isoforms T, P and C were analyzed in these experiments. The T isoform (also designated as 67.2) represents the most extensively characterized Mod(mdg4) product, which functions along with Su(Hw) and CP190 in the gypsy insulator for example [37]. The C and P isoforms might be expressed in testis according to RNA-seq data [31]. In addition, we used a construct for expression of the shared N-terminal common part (CP) of the Mod(mdg4) proteins. Our experiments revealed that none of the analyzed Mod(mdg4) variants (C, P, T and CP, all with C-terminal mCherry) were able to co-immunoprecipitate SNM-EGFP and UNO-myc, in contrast to MNM-mCherry (Fig 2B). We conclude that the isoform-specific C-terminal region of MNM (MNM_C) is required for binding to the SU complex. Additional experiments confirmed that MNM_C fused to mCherry (MNM_C-mCherry) is co-immunoprecipitated by UNO-EGFP after co-expression with SNM (Fig 2G).

The N-terminal common part of the Mod(mdg4) proteins includes a BTB/POZ domain, which is present in a wide range of proteins with distinct functions (Fig 2A). BTB/POZ domains appear to mediate protein-protein interactions, including dimerization, tetramerization and multimerization [38,39]. Therefore, we evaluated whether MNM self-associates via the N-terminal common part (CP) and hence also with Mod(mdg4) proteins other than MNM. Indeed, MNM-EGFP was readily co-immunoprecipitated with all the analyzed Mod (mdg4) variants C, P, T and CP fused to mCherry (Fig 2C). Similarly, we detected co-immuno-precipitation of MNM-EGFP and MNM-myc (Fig 2D). We conclude that MNM has the potential to associate with itself and with other Mod(mdg4) proteins via the N-terminal common part. The isoform-specific C-terminal region of MNM fused to mCherry (MNM_C-mCherry) was not observed to co-immunoprecipitate with MNM-EGFP (Fig 2E).

The isoform-specific C-terminal region of MNM (MNM_C) does not only bind to the SU complex (Fig 2A and 2B) but also to the N-terminal region of TEF [23]. Moreover, consistent with MNM's ability to bind to both the SU complex and to TEF, all four AHC proteins (SNM, UNO, MNM and TEF) are co-immunoprecipitated after co-expression in S2R+ cells (Fig 2F) [23]. This apparent SUMT complex could arise if the binding of SU and TEF to MNM was not mutually exclusive. However, even if mutually exclusive, SUMT complex formation might still succeed, because of the self-association of MNM. For example, one of two associated MNM proteins might bind to SU and the other to TEF. To address the mode of TEF binding in the SUMT complex, we co-expressed TEF-myc with MNM_C-mCherry, SNM and UNO_-C-EGFP (Fig 2G). As shown above, MNM_C cannot self-associate, but it can bind to TEF and also to the SU complex. UNO_C cannot self-associate, but it can bind to SNM and recruit MNM_C. As expected, UNO_C-EGFP was observed to co-immunoprecipitate MNM_C-mCherry after co-expression of the four proteins (TEF-myc, MNM_C-mCherry, SNM and UNO_C-EGFP), but TEF-myc was not co-immunoprecipitated (Fig 2G). Thus, after binding to the SU_C-EGFP complex, MNM_C-mCherry was no longer able to bind TEF-myc. This result suggests that the binding of TEF and SU to MNM is mutually exclusive. Accordingly,

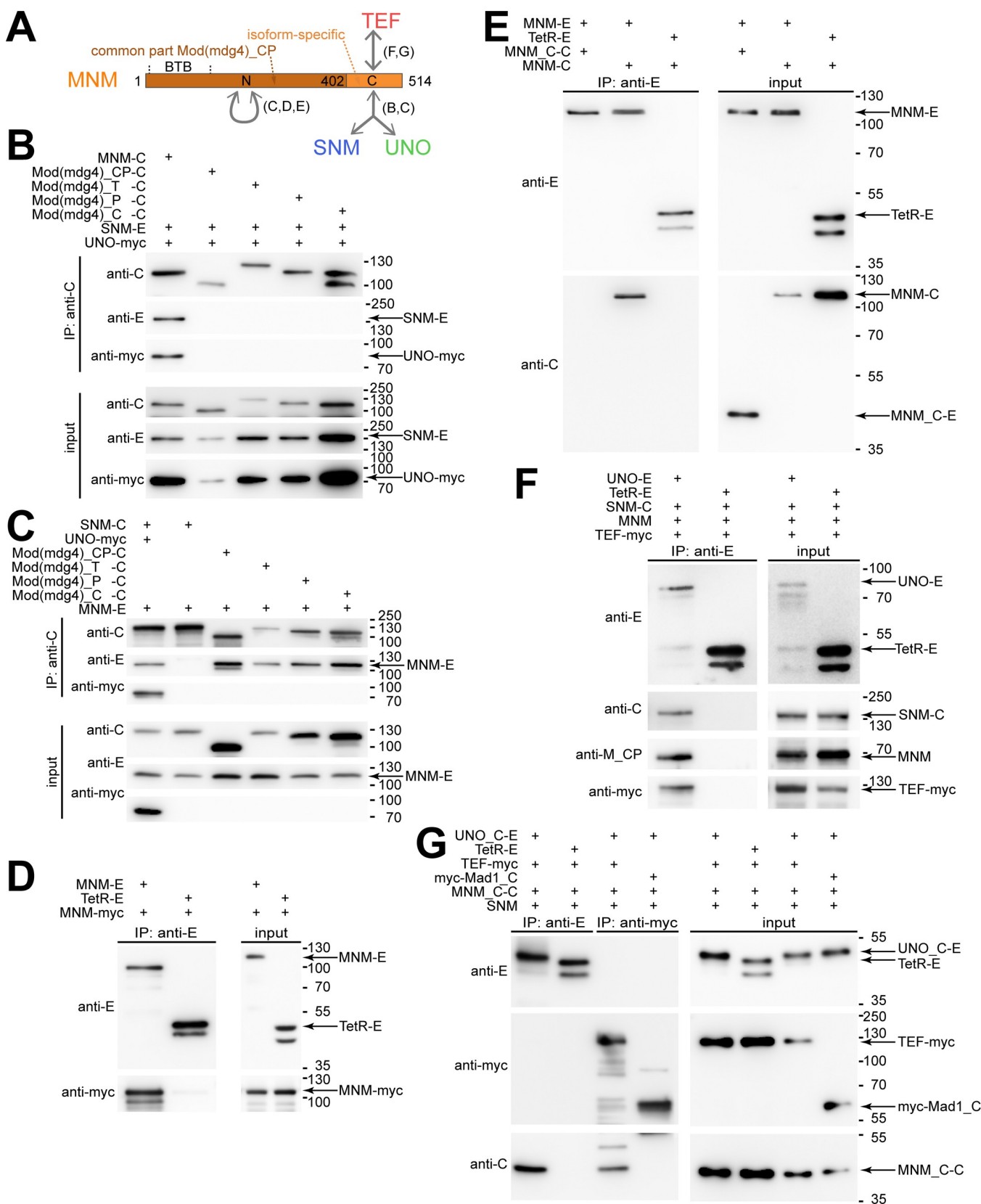

**Fig 2. MNM self-association is required for formation of SUMT complexes containing SNM, UNO, MNM and TEF.** (**A**) Structure of MNM and interactions with AHC proteins. MNM has an N-terminal (Mod(mdg4)_CP) that is also present in other Mod(mdg4) isoforms (including _T, _P and _C). This part includes a BTB domain. In addition, MNM has a C-terminal MNM-specific region. Protein interactions were analyzed by co-immunoprecipitation experiments after transient expression in S2R+ cells. The main findings are summarized schematically (with reference to relevant figure panels in brackets). (**B-G**) Co-immunoprecipitation after transient expression of the indicated proteins was analyzed by immunoblotting. Protein tags were EGFP (E), mCherry (C) or a myc epitope (myc). Tetracycline repressor fused to a nuclear localization signal and EGFP (TetR-E) or a C-terminal fragment of Mad1 tagged with a myc-epitope (myc-Mad1_C) was used for control experiments. Presence or absence of proteins in the extracts used for immunoprecipitation (input) or in the samples (IP) immunoprecipitated with the indicated antibodies were analyzed with anti-EGFP (anti-E), anti-mCherry (anti-C), anti-myc (anti-myc), and anti-Mod(mdg4)_CP (anti-M_CP). The positions of molecular weight markers are indicated on the right side, as well as the bands representing the indicated proteins of interest. (**B**) While MNM-mCherry (MNM-C) co-immunoprecipitated SNM-EGFP (SNM-E) and UNO-myc (UNO-myc), other Mod(mdg4) isoforms (T, P and C) as well as the common part (CP) tagged with mCherry did not co-immunoprecipitate SNM-EGFP and UNO-myc. (**C**) The mCherry fusions of Mod(mdg4) variants (CP, T, P and C) all co-immunoprecipitated MNM-EGFP (MNM-E) specifically, as demonstrated by first two lanes, which present a positive and negative control experiment, respectively. (**D**) MNM-E co-immunoprecipitated MNM-myc specifically, as demonstrated by the negative control experiment with TetR-E. (**E**) The C-terminal MNM-specific region as an mCherry fusion (MNM_C-C) was not co-immunoprecipitated by MNM-E in contrast to full length MNM-mCherry (MNM-C). (**F**) UNO-E co-immunoprecipitated SNM-mCherry, MNM and TEF-myc specifically [23], as demonstrated by the negative control experiment with TetR-E. (**G**) UNO_C-E co-immunoprecipitated only SNM-mCherry, MNM_C-C but not TEF-myc, even though TEF-myc was observed to co-immunoprecipitate MNM_C-C, indicating that MNM_C can bind in a mutually exclusive manner to either the SNM-UNO complex or to TEF.

formation of SUMT assemblies, which contain all four known AHC proteins, depends on self-association of either MNM or possibly also of UNO.

## Purification and structural analysis of a complex of SNM and UNO_C

For further characterization of the AHC protein-protein interactions, we succeeded in purifying recombinant versions of some of these proteins for analysis *in vitro*. To characterize the SU complex, we expressed full-length SNM and UNO_C (aa 281–417) in baculovirus-infected insect cells. To promote solubility, SNM was fused N-terminally with a maltose-binding protein (MBP). UNO_C was fused to an N-terminal twin Strep-II affinity tag. Using streptactin affinity chromatography followed by size exclusion chromatography, we purified an MBP-SNM/Strep-II-UNO_C complex to homogeneity (from here on referred to as SU_C) (Fig 3A). We next determined the stoichiometry of SU_C using both mass photometry (MP) and multi-angle light scattering coupled to size exclusion chromatography (SEC-MALS). MP and SEC-MALS measured a molecular mass of 178 and 211.9 kDa, respectively (Fig 3B and 3C). As the theoretical molecular mass of a 1:1 dimer was 173.4 kDa, these results provided strong evidence of a 1:1 stoichiometry. Based on this stoichiometry, we generated a *de novo* structure prediction using AlphaFold 2.2.0 Multimer (AF2) [40,41] (Fig 3D). In order to test the AF2 model, we made use of bifunctional chemical cross-linking coupled to mass spectrometry (XL-MS). The observed cross-links were mapped onto the 2D representation of the complex and onto the 3D model (Fig 3E). Given that Disuccinimidyl Dibutyric Urea (DSBU), which was used as cross-linker, has a fixed length (12.5 Å), we would expect the observed cross-links to occur between residues that are no further apart in the 3D model than the maximum possible cross-link Cα-Cα distance. The maximum distance is usually considered to be linker length plus 2 x lysine length (12.8 Å), and an additional tolerance of between 2-5Å [42]. The majority of the detected cross-links mapped onto the AF2 model within an appropriate distance limit (i.e. <30Å) (Fig 3E), thus suggesting that the SU_C structure model is likely a good fit.

## DNA-binding of the SNM-UNO_C complex

The predicted SU_C structure displayed striking similarity to the structures reported for complexes of kleisins bound to HAWK-type subunits of both cohesin and condensin complexes [2]. The SU_C structure was particularly similar to that of the human cohesin subunits STAG1 and RAD21 [36] (Fig 4A). Like SNM, STAG1 belongs to the stromalin family, and RAD21 is

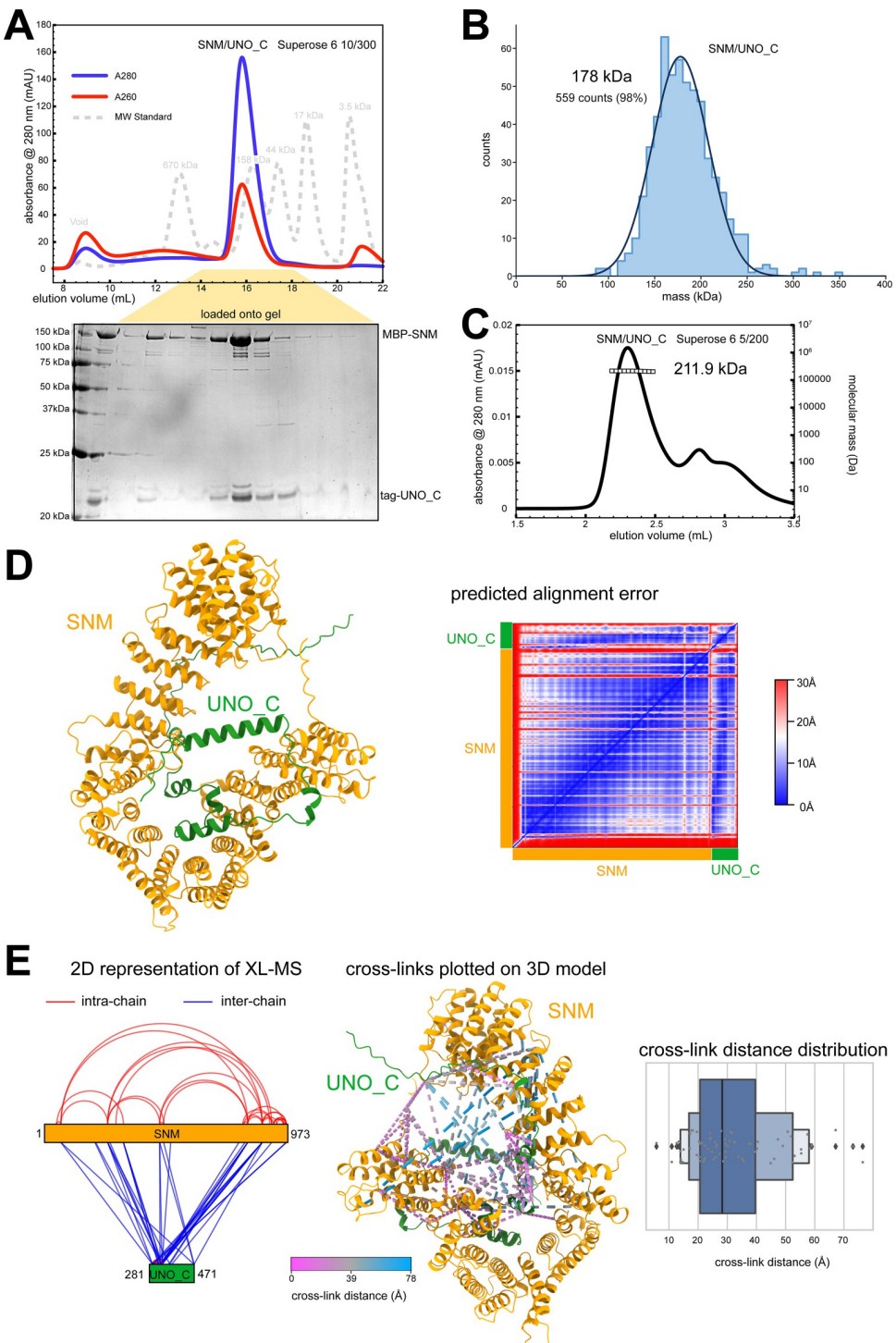

**Fig 3. SNM and UNO_C form a stable heterodimeric complex.** (**A**) SNM and UNO_C, with N-terminally fused maltose-binding protein (MBP) and twin Strep-II affinity tag (Stag), respectively, form a stable SU_C complex. Size exclusion chromatography profile from the final purification step and Coomassie-stained gel for analysis of the indicated peak are shown. (**B**) Mass photometry analysis of SU_C. The indicated molecular mass was determined by a Gaussian fit of the distribution of counts calibrated against a molecular mass standard. (**C**) SEC-MALS analysis of SU_C, revealing the indicated molecular mass. (**D**) AF2 model of SU_C with predicted alignment plot on the right. Regions associated with high (30 Å) and low (0 Å) error as predicted by the algorithm are shown in red and blue, respectively. (**E**) Analysis of SU_C by cross-linking mass spectrometry (XL-MS). The observed cross-links (with false discovery rate <1%) were plotted onto 2D representations of SNM and UNO_C. Moreover, cross-links were also

modelled onto the AF2 model using XMAS [74], with separation distance of cross-linked positions color-coded. A plot with the distance distribution of the observed cross-links mapped to the AF2 model is presented on the right. The majority of cross-links fall within the distance range expected for the DSBU linker (~27 Å, see text and [42]. The longer distance outliers may be caused by errors in the model prediction, or flexibility within the structure.

an α-kleisin. The similarity between SU_C and STAG1/RAD21 included the apparent conservation of three surface patches (P1-P3) with positively charged residues that are involved in the binding of cohesin to DNA [43] (Fig 4B). The regions corresponding to P1 and P3 in particular were found to be positively charged. Interestingly, compared to RAD21, UNO has an extra α-helix at the C-terminus (Fig 4B). This C-terminal helix (ch) is highly positively charged and its position within the AF2 model was also suggestive that it might cooperate with the basic patches for DNA-binding (Fig 4B).

To evaluate whether SU_C might indeed bind to DNA, we performed electrophoretic mobility shift assays (EMSAs). Thereby, SU_C was observed to have DNA-binding activity (Fig 4C) that appeared comparable or even higher than that reported for human cohesin [43]. In spermatocytes, SNM, UNO and MNM are all co-localized on the sex chromosome pairing regions, i.e., on the rDNA loci [20,24–26]. Specifically, the 240 bp repeats located within the intergenic spacers of the rDNA repeats were shown to be sufficient for chromosome conjunction. Therefore, we tested whether SU_C might bind preferentially to the 240 bp repeat sequence with an EMSA competition experiment. We mixed two DNA sequences, one corresponding to the 240 bp repeat sequence, and a second with the same sequence composition but scrambled. The former was labelled with Cy5 and the latter with fluorescein. Both sequences were bound equally well by SU_C (Fig 4C). Thus, SU_C does not appear to have a binding preference for the 240bp repeat sequence.

In order to test the potential involvement of the basic patches (P1, P2 and P3) of SNM and the C-terminal helix (ch) of UNO in DNA-binding, we generated mutants. The basic patches were altered with a series of point mutations analogous to the work on STAG1/RAD21 [43], and were designated as SNM$^{P1m}$, SNM$^{P2m}$ and SNM$^{P3m}$. In case of the C-terminal helix of UNO, we generated the mutant UNO$^{chm}$ with six basic residues changed into either alanine of glutamic acid (R395E, K397A, R398E, R401A, R405E, and R406A). During purification of these four mutant SU_C complexes, SNM$^{P3m}$ proved to be unstable and was therefore not pursued further. The remaining three mutant complexes could be purified to homogeneity like wild-type SU_C (Fig 4D). Compared to U_C, the mutant U_C$^{chm}$ had a lower mobility during SDS-PAGE (Fig 4D). As SNM$^{P2m}$ did not appear to bind stoichiometric amounts of UNO_C (Fig 4D, asterisk), we also did not analyze it further. However, we compared the efficiency of DNA-binding of the three complexes wild-type SU_C, S$^{P1m}$U_C and SU_C$^{chm}$ with EMSAs. Wild-type SU_C bound DNA with a high affinity ($K_D$ of 555 nM +/- 37) (Fig 4E). S$^{P1m}$U_C bound with an approximate three-fold lower affinity. SU_C$^{chm}$ bound with an even lower affinity (Fig 4E). These results demonstrate that SU_C binds efficiently to DNA. Moreover, they implicate the characteristic highly basic C-terminal helix of UNO in DNA-binding.

## The positively charged C-terminal helix of UNO promotes chromatin binding and chromosome conjunction

To further evaluate the role of the strongly positively charged C-terminal helix of UNO, we generated an *UASt-uno$^{chm}$-EGFP* transgene for analyses *in vivo*. Wild-type UNO was previously shown to bind to polytene chromosomes in larval salivary gland after co-expression with SNM [23]. To determine whether UNO$^{chm}$ can still bind to polytene chromosomes, we co-expressed *UASt-snm-mCherry* and *UASt-uno$^{chm}$-EGFP* with the salivary gland-specific driver

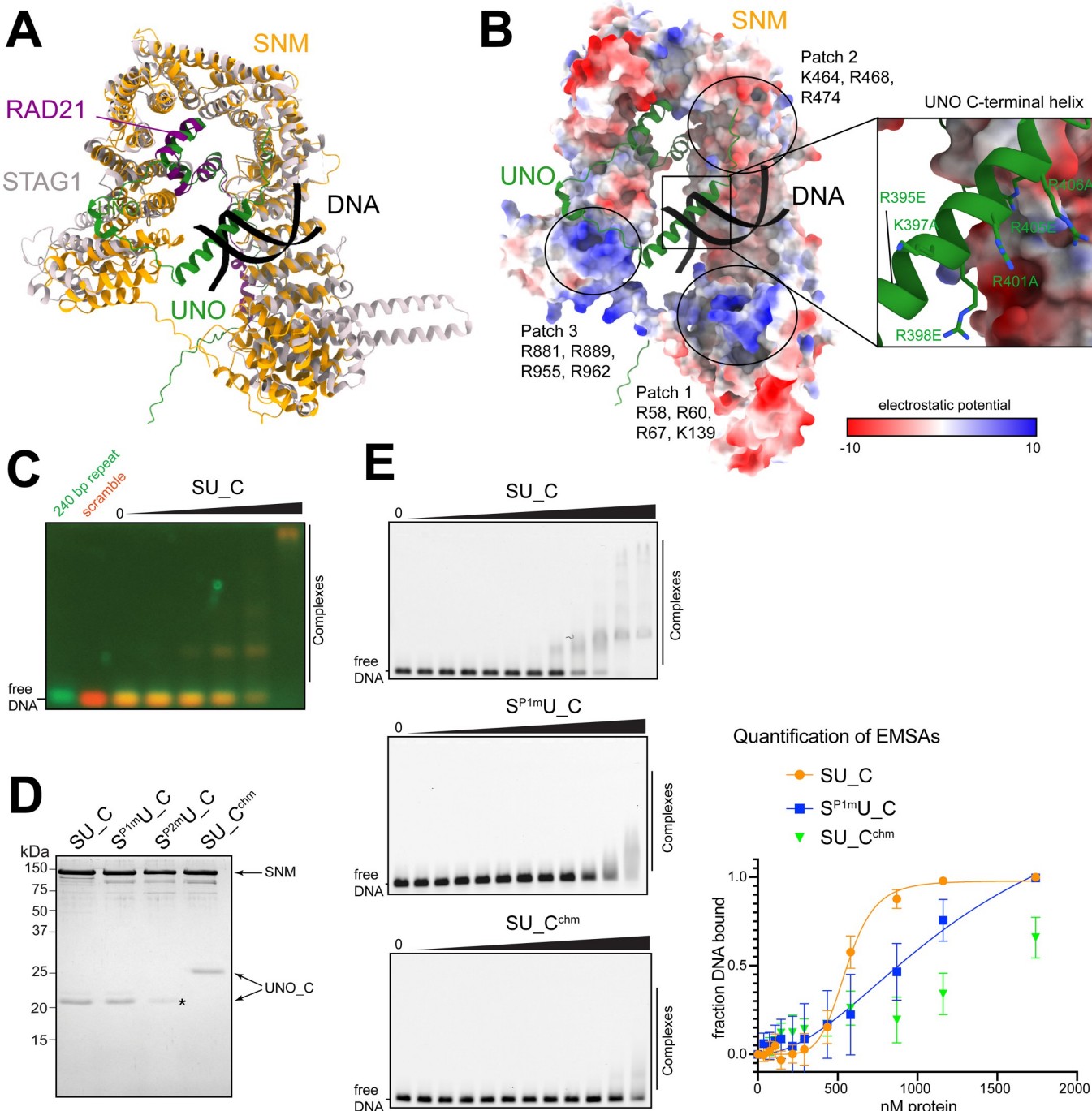

**Fig 4. The SU_C complex binds to DNA.** (**A**) Structural comparison of SU_C with STAG1/RAD21 from the human cohesin complex. The structure of STAG1, RAD21 and dsDNA (PDB 6wg3, [36]) was superposed on the SU_C model using Chimera MatchMaker, giving a local C-alpha RMSD of 1.31 Å over 272 residues, and a global C-alpha RMSD of 8.1 Å over all 863 pairs. (**B**) Surface representation of SNM colored by electrostatic potential to highlight three potential DNA-binding patches that are common with STAG1. UNO_C is shown as a cartoon representation. The inset (right) shows the positively charged residues of the C-terminal helix of UNO that were mutated (see below). The position of dsDNA is taken from the STAG1/RAD21 model shown in (A). (**C**) DNA-binding of SU_C. For analysis by EMSA, SU_C was incubated with a mix of two distinct DNA fragments, fluorescently labeled with a green and a red dye, respectively. The sequence of the first DNA fragment corresponded to that of the 240 bp repeat previously implicated in sex chromosome conjunction before male M I [26]. A scrambled version of the same sequence was present in the second fragment. Band shifts revealed DNA-binding by SU_C without preference for the 240 bp repeat. (**D**) SU_C and mutant versions of this complex were purified and analyzed by SDS-PAGE and Coomassie staining. S$^{P1m}$: SNM with mutant basic patch 1, S$^{P2m}$: SNM with mutant basic patch 2, C$^{chm}$: UNO_C with mutations in C-terminal helix. S$^{P2m}$ resulted in a reduced presence of UNO_C (asterisk) and C$^{chm}$ in an altered electrophoretic mobility of UNO-C. (**E**) Representative EMSAs for comparison of DNA-binding activity of SU_C

complex variants (wild-type SU_C, $S^{P1m}U$_C, and SU_C$^{chm}$). The range of the analyzed protein concentrations was from 36 nM to 1.7 µM in each experiment. Quantification of the EMSA experiments is shown on the right. Three independent experiments were carried out and the fraction of bound DNA at each data point was calculated. Error bars show s.d. A non-linear regression fit was made to determine apparent $K_D$. A good fit (R-squared 0.955) was obtained for the wild-type SU_C dataset, resulting in a $K_D$ of 555 nM (+/- 37), with a hill-factor of 6. For the mutant $S^{P1m}U$_C, the fit was less good, with an approximate apparent $K_D$ of 1.3 µM. For the mutant SU_C$^{chm}$, no curve could be fitted.

*Sgs3-GAL4.* In spread polytene chromosome preparations, the yellow signals in chromosomal bands resulting from strictly co-localized SNM-mCherry and UNO$^{chm}$-EGFP were strongly reduced in intensity compared to controls co-expressing SNM-mCherry and wild-type UNO-EGFP (Fig 5A). The reduced polytene chromosome-binding resulting from co-expression of SNM-mCherry with UNO$^{chm}$-EGFP did not arise because of lower expression levels, as demonstrated by quantification of fluorescent signals in whole mount preparations (Fig 5C). On the contrary, co-expression of SNM-mCherry with UNO$^{chm}$-EGFP resulted in higher expression levels compared to controls with SNM-mCherry and wild-type UNO-EGFP (Fig 5C). However, the combination of SNM-mCherry with UNO$^{chm}$-EGFP was localized predominantly in between polytene chromosomes (Fig 5B), in contrast to the combination of SNM-mCherry with wild-type UNO-EGFP that was preferentially on polytene chromosomes (Fig 5B).

To confirm that UNO$^{chm}$ still binds to SNM and MNM, we performed co-immunoprecipitation experiments after transient co-expression in S2R+ cells. The results clearly confirmed that UNO$^{chm}$-EGFP still binds to SNM-mCherry and MNM (Fig 5D). Moreover, the purification of recombinant SU_C complexes (Fig 4D) had also demonstrated normal SNM binding by UNO$^{chm}$_C.

According to the results of our analyses in larval salivary glands, in S2R+ cells and *in vitro*, the mutations in UNO$^{chm}$, which eliminate positive charge from the C-terminal helix, did not interfere with SNM and MNM binding but they strongly reduced chromosomal recruitment. To analyze UNO$^{chm}$ function during male meiosis, we expressed *UASt-uno$^{chm}$-EGFP* in *uno* null mutants using *bamP-GAL4-VP16*. Analogous expression of wild-type *UASt-uno-EGFP* in *uno* null mutants was shown to preclude premature disjunction and random segregation of homologs during M I [22]. In contrast, meiotic chromosome segregation was clearly defective in case of *bam> uno$^{chm}$-EGFP* in *uno* null mutants. This was revealed by measuring the nuclear DNA content in individual post-meiotic nuclei of early spermatids. Regular chromosome segregation during wild-type meiosis generates a population of haploid nuclei with highly comparable DNA content, while random chromosome segregation during M I in *uno* null mutants generates post-meiotic nuclei with a highly variable DNA content [22]. In case of *bam>uno$^{chm}$-EGFP* in *uno* null mutants, early spermatids displayed an extent of variability of the nuclear DNA content that was intermediate between that in wild-type controls and *uno* null mutants (Fig 5E) [22]. Therefore, UNO$^{chm}$-EGFP does not restore normal meiotic chromosome segregation. Meiotic chromosome segregation was also not entirely normal when *UASt-uno$^{chm}$-EGFP* was expressed with *bamP-GAL4-VP16* in heterozygous *uno$^{cc1}$* spermatocytes (Fig 5E), presumably reflecting a dominant-negative effect of UNO$^{chm}$-EGFP.

For further functional characterization of UNO$^{chm}$-EGFP, we analyzed its localization during spermatogenesis. UNO$^{chm}$-EGFP expression was readily detectable in squash preparations of testes from *uno* null mutants with *bam> uno$^{chm}$-EGFP*. Beginning at around the S3 stage, the subcellular localization of UNO$^{chm}$-EGFP diverged from that of wild-type UNO-EGFP. In contrast to wild-type UNO-EGFP, which relocated from a diffuse nucleolar distribution into sub-nucleolar foci, UNO$^{chm}$-EGFP signals maintained a homogenous distribution in the nucleolus (Fig 5F). Eventually, in S5 spermatocytes, UNO$^{chm}$-EGFP was enriched around rather than within the nucleolus (Fig 5F). Beyond these diffuse perinucleolar UNO$^{chm}$-EGFP

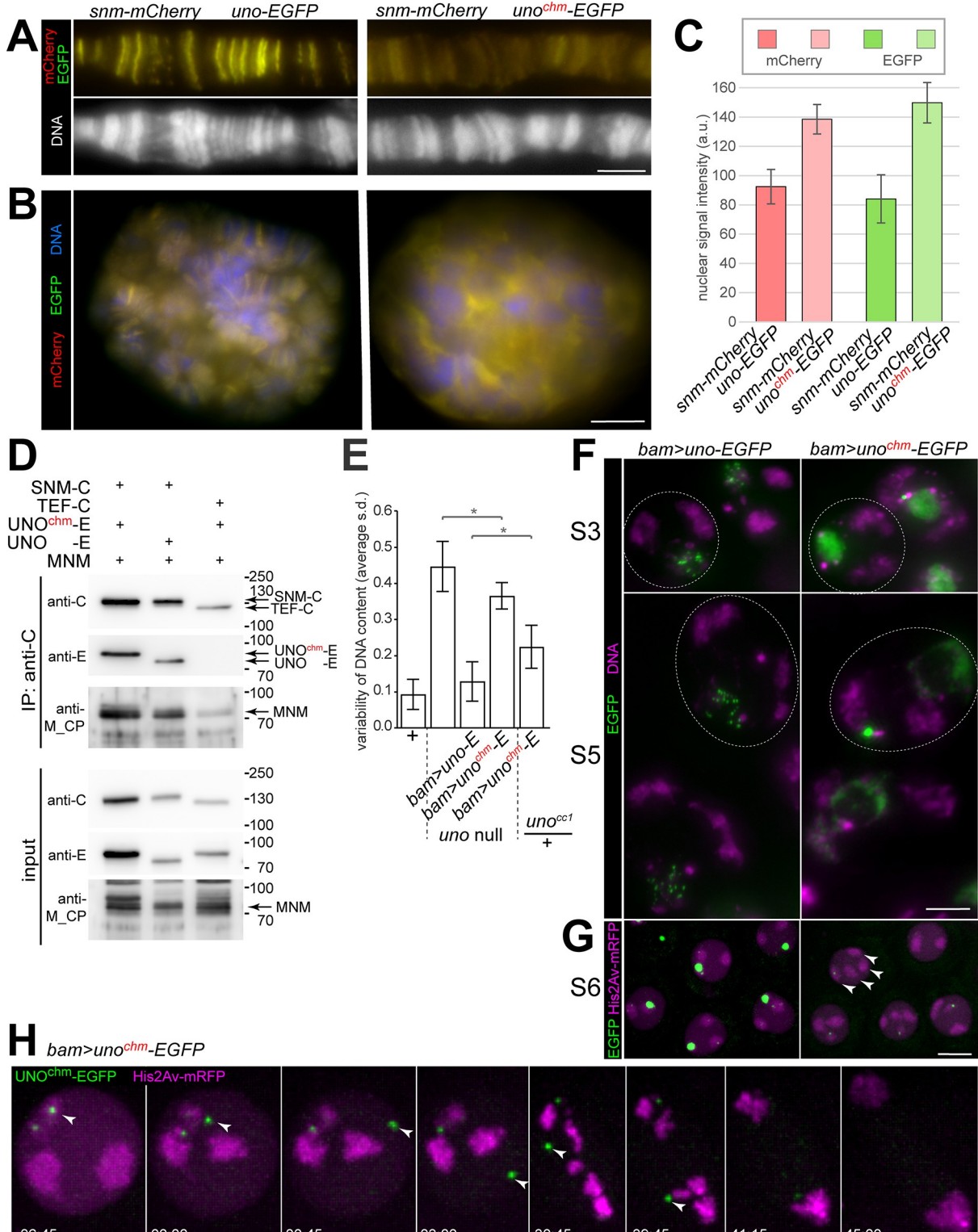

**Fig 5. Mutations eliminating positive charge from the C-terminal helix of UNO decrease chromatin binding of SU and interfere with normal chromosome conjunction during meiosis.** (**A**) After co-expression with SNM-mCherry in larval salivary glands, UNO^chm^-EGFP with a mutant C-terminal helix is still strictly co-localized with SNM-mCherry, as indicated by yellow signals, but its association with polytene chromosome spreads is much weaker in comparison to wild-type UNO-EGFP. *UASt* transgenes and *Sgs3-GAL4* were used for salivary gland-specific expression. (**B,C**) Expression of SNM-mCherry and UNO^chm^-EGFP resulting with the *UASt* transgenes and *Sgs3-GAL4* was stronger

than the analogous expression of SNM-mCherry and wild-type UNO-EGFP, as revealed by microscopic analysis of whole mount preparations of larval salivary glands. However, the former combination (with UNO^chm-EGFP) resulted in signals predominantly in between polytene chromosomes in contrast to the latter combination (with wild-type UNO-EGFP), which was primarily chromosome-associated, as revealed by the optical slices of nuclei. The bar diagram (C) displays average intensities of the mCherry and EGFP signals in the salivary gland nuclei expressing the indicated transgenes, as well as s.d., n = 6 glands (UNO-EGFP) and 7 glands (UNO^chm-EGFP). (**D**) UNO^chm-EGFP and wild-type UNO-EGFP are co-immunoprecipitated to a comparable degree along with MNM by SNM-mCherry. For further explanations, see legend of Fig 1. (**E-H**) Function of UNO^chm-EGFP in spermatocytes. The driver *bamP-GAL4-VP16* was used for expression of either *UASt-uno^chm-EGFP* or *UASt-uno-EGFP* for comparison. (**E**) Extent of chromosome missegregation during meiosis in the indicated genotypes was estimated by analysis of the variability of the DNA content detected microscopically in nuclei of early round spermatids in testis squash preparations. Bars indicate the average of the standard deviation of the DNA content distribution observed in distinct spermatid cysts, and whiskers indicate s.d. of these averages. The first two bars on the left represent data reported previously [22]. (**F**) Squash preparations of *uno* null mutant testes with either *bam> uno-EGFP* or *bam>uno^chm-EGFP* were labeled with a DNA stain. Representative spermatocytes at the indicated stages reveal an abnormal UNO^chm-EGFP localization in the nuclei (dashed circumference). (**G**) Spermatocytes homozygous for the *uno*^cc1 null mutation with *His2Av-mRFP* and either *bam>uno-EGFP* or *bam>uno^chm-EGFP* were analyzed by time-lapse imaging. UNO^chm-EGFP dot signals were far weaker than those formed by UNO^chm-EGFP on the sex chromosome bivalent in spermatocytes from cysts at the S6 stage just before NEBD I. Moreover, the number of major chromosome territories (arrowheads) was increased above the normal number of three in the spermatocytes expressing UNO^chm-EGFP instead of endogenous UNO. (**H**) Progression through M I in spermatocytes homozygous for the *uno*^cc1 null mutation with *His2Av-mRFP* and *bam>uno^chm-EGFP* was analyzed by time-lapse imaging. An UNO^chm-EGFP dot moving very rapidly without an associated chromosome mass is marked by an arrowhead until its abrupt disappearance during anaphase I. Premature separation of bivalents before the onset of anaphase I is apparent at the time points 30:45 and 39:45 (min:sec after onset of NEBD I). Scale bars = 5, 10, 5, 10 and 2 μm in A, B, F, G and H, respectively.

signals, some late spermatocytes displayed also one or two strong EGFP dots in the nucleus. In contrast, wild-type UNO-EGFP was still predominantly in strong intranucleolar foci during the S5 stage (Fig 5F).

To characterize chromosome segregation during M I in *uno* null mutants with *bam> uno^chm-EGFP*, we applied time-lapse imaging. As reported earlier for controls (*uno* null mutants with *bam> uno-EGFP*), a coalescence of sub-nucleolar UNO-EGFP foci into a strong dot on the pairing center of the chrXY bivalent occurs at the end of the S6 stage (Fig 5G) [22]. In comparison, the UNO^chm-EGFP dot signals in late S6 spermatocytes were weaker and more variable in *uno* null mutants with *bam> uno^chm-EGFP* (Fig 5G). Regarding the number of major chromosome territories, most spermatocytes in *uno* null mutants with *bam> uno^chm-EGFP* appeared to be still normal at the onset of M I. Ninety two percent of the spermatocytes displayed three major territories, as in controls (n = 66 from seven distinct cysts). The remaining 8% had four completely separated territories (Fig 5G). However, during prometaphase I, 94% of the spermatocytes displayed premature separation of major bivalents, followed by random segregation of univalents during anaphase I (Fig 5H and S1 Movie). Only four spermatocytes (6%) progressed through M I normally. Unexpectedly, 17% of the spermatocytes analyzed, displayed an EGFP dot with aberrant characteristics. These EGFP dots were not linked to chromosomal His2Av-mRFP masses, and their movements were far more rapidly than those of chromosomes (Fig 5H and S1 Movie). During anaphase I, these EGFP dot signals, as well as those associated with a chromosome disappeared rapidly (Fig 5H and S1 Movie), like wild-type UNO-EGFP [22].

In conclusion, our characterization of UNO^chm-EGFP function demonstrated the importance of the C-terminal helix of UNO for normal male meiosis. This conserved helix is characteristic for UNO orthologs and absent from α-kleisins. Consistent with the observations concerning DNA-binding *in vitro*, mutational elimination of the positive charge in this helix impaired binding to chromosomes and homolog conjunction in spermatocytes.

## Multimerization by N-terminal domains of UNO and MNM

The N-terminal part of UNO, which is most conserved but unrelated to α-kleisins, permits self-association according to our co-immunoprecipitation experiments. For further structural characterization, we purified recombinant UNO_N (aa 1–73). As for SNM, UNO_N was

made more soluble with an N-terminal MBP fusion. MBP-UNO_N was purified using amylose affinity chromatography, ion-exchange chromatography and ultimately size exclusion chromatography (SEC). Although MBP-UNO_N was largely free of contaminants, it was present in two peaks during the final SEC (S4A Fig). We kept these peaks separate for measurement of molecular masses using SEC-MALS. Peak 1 and 2 had a mass of 200.2 and of 101.2 kDa, respectively (S4B Fig). These data suggested that peak 1 is a tetramer (theoretical molecular mass 204.56 kDa) and peak 2 a dimer (theoretical molecular mass 102.28 kDa) of UNO_N. We again made use of AF2 modelling using the knowledge of the two UNO_N stoichiometries in our input parameters. Both UNO_N dimer and tetramer generated high confidence models (S4 Fig). These results confirm that UNO can self-associate and indicate that it can form dimers and/or tetramers.

Our co-immunoprecipitation experiments revealed self-association not only for UNO_N but also in case of Mod(mdg4)_CP, the common N-terminal part shared between MNM and the other Mod(mdg4) protein isoforms. For further structural characterization, we expressed Mod(mdg4)_CP in *E. coli* with an N-terminal MBP moiety to promote solubility. After affinity purification using amylose beads, MBP was removed with 3C protease. Ultimately, the protein was purified to homogeneity using SEC (S5A Fig). Strikingly, the elution volume of Mod(mdg4)_CP was considerably larger than expected (S5A Fig). Thus, we measured the molecular mass using SEC-MALS (S5B Fig). The resulting value of 80.92 kDa was a nearly perfect match for a hexamer of Mod(mdg4)_CP (theoretical molecular mass of 81.76 kDa). Mod(mdg4)_CP contains a BTB/POZ motif, for which self-association has been extensively reported [38,39], although never as hexamer. We therefore used AlphaFold 2.2.0 multimer to generate structural models of the Mod(mdg4)_CP hexamer, which suggested a ring-like arrangement of the subunits (S5 Fig). For confirmation of the hexameric ring architecture, we carried out negative stain electron microscopy (NS-EM), which revealed small ring-like assemblies of Mod(mdg4)_CP with a diameter consistent with the AlphaFold model (S5 Fig).

## SUM complexes on the sex chromosome pairing region are stable

To evaluate whether the nucleolar SUM foci in spermatocytes are temporally stable or dynamic, we used fluorescence recovery after photobleaching (FRAP) for analysis. Stable foci are expected, if the physical linkage of chromosomes into bivalents is achieved by SUM protein complexes functioning like a conventional glue. However, in principle, homologs might also be conjoined by temporally dynamic protein assemblies, if some of many remain in place at any given time point, for instance as in liquid-liquid phase separated droplets. The FRAP analysis was done with S5 spermatocytes expressing either SNM-EGFP, UNO-EGFP or MNM-EGFP (using *bamP-GAL4-VP16* and *UASt* transgenes). Because autosomal signals were too weak for analysis, we focused on the intense nucleolar signals. A region comprising about half of the nucleolus was bleached rapidly (Figs 6A and S6). We obtained comparable results from eight cells per genotype and each cell was from a distinct cyst. For analysis of FRAP, we quantified EGFP signal intensities and used the EGFP signals in nucleoli of neighboring spermatocytes, which were not bleached as reference (S6 Fig). Recovery of EGFP signals in the bleached region was observed to be slow and partial (Figs 6A, 6B, and S6). After 90 minutes, recovery of the EGFP signal intensities was maximal in case of SNM-EGFP (50%), intermediate for UNO-EGFP (29%) and minimal for MNM-EGFP (14%) (Fig 6B). However, recovery was restricted to the weaker diffuse nucleolar signals around the intense sub-nucleolar foci. Signals in the sub-nucleolar foci, which remained in a stable spatial pattern in neighboring non-bleached control spermatocytes, did not recover in the bleached nucleolar regions. For further confirmation, we analyzed MNM-EGFP recovery 240 minutes after photobleaching,

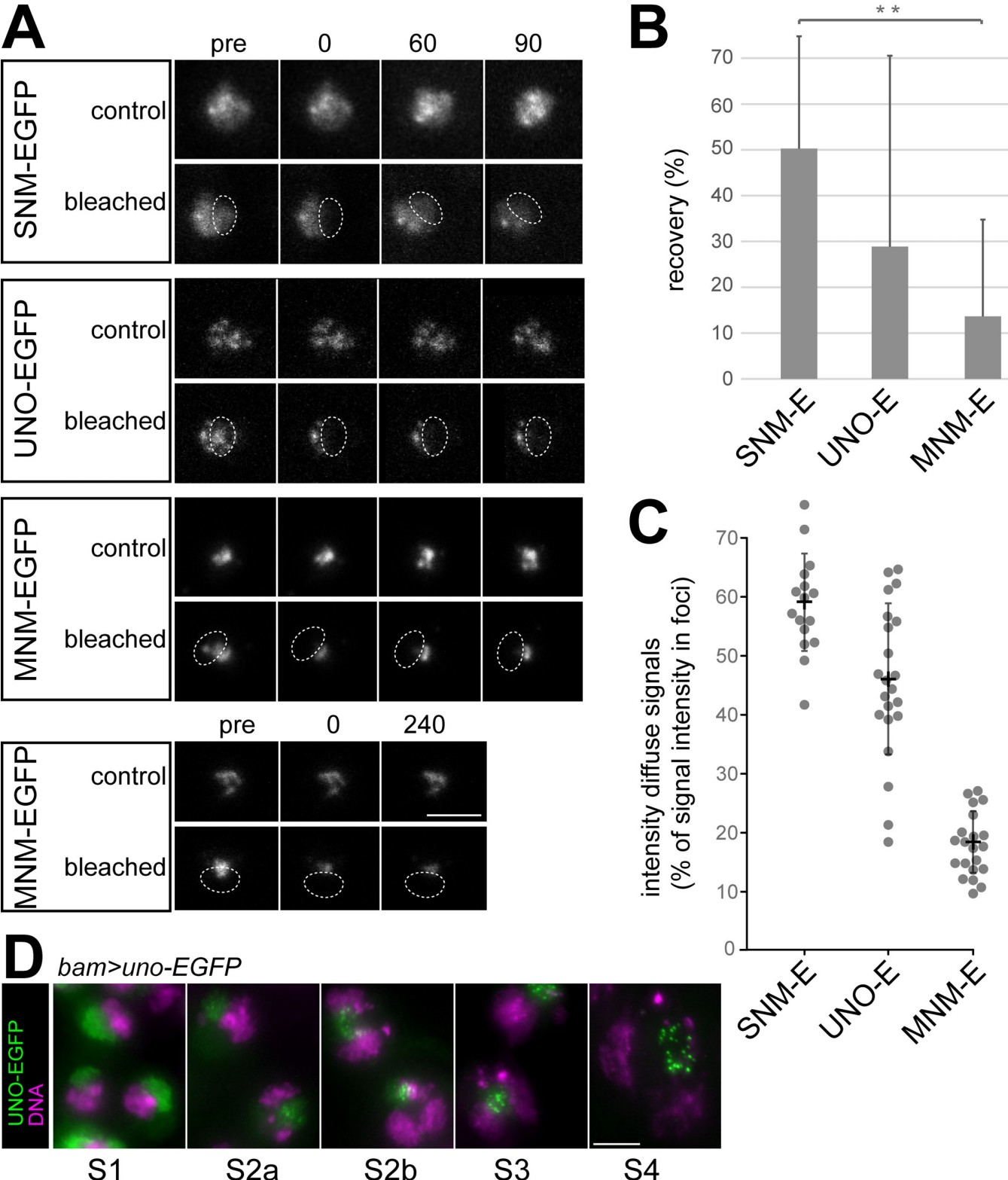

**Fig 6. SUM complexes on the sex chromosome pairing region are stable.** (**A**) FRAP analyses were completed with S5 cysts released from pupal testes of males expressing the indicated *UASt* transgenes driven by *bam-GAL4-VP16*. A subregion of the nucleolus (dashed oval) was bleached in one of the spermatocytes at t = 0. Still frames from representative control and bleached spermatocytes are displayed immediately before bleaching (pre) and at the indicated time points (min). (**B**) The extent of fluorescent signal recovery at 90 minutes after photobleaching was quantified. Bars indicate average and

whiskers s.d.; n = 8 spermatocytes per genotype. (**C**) Intensities of the EGFP signals in sub-nucleolar foci and of the surrounding diffuse nucleolar signals were quantified during stage S5 (late). Dots indicate diffuse signal intensities relative to the intensities in sub-nucleolar foci, which were set as 100%. Each dot represents a distinct spermatocyte. Averages +/- s.d. are indicated as well; n = 16, 22 and 21 spermatocytes (from left to right). (**D**) Transition of the subcellular localization of UNO-EGFP in nucleoli during spermatocyte maturation (S1 to S4 stage) from diffuse to sub-nucleolar foci as observed in squash preparations of *bam>uno-EGFP* testes labeled with a DNA stain. Scale bars = 5 μm.

and again did not detect any recovery of the signals in the sub-nucleolar foci in three independent experiments (Fig 6A).

The extent of recovery of the weak diffuse nucleolar signals that was observed for the different AHC proteins was correlated with the distinct levels of these diffuse signals before photobleaching (Fig 6C). Compared to the average pixel intensity in the sub-nucleolar foci, the diffuse nucleolar signals were highest in case of SNM-EGFP (60%), intermediate for UNO-EGFP (48%) and lowest for MNM-EGFP (18%), when analyzed at the S5 stage, which was also studied in the FRAP experiments (Fig 6C). Interestingly, the diffuse nucleolar signals were dominant in early spermatocytes, and sub-nucleolar foci developed during spermatocyte maturation, as illustrated for UNO-EGFP (Fig 6D) and reported earlier for MNM- and SNM-EGFP [44]. Overall, the results of our FRAP experiments indicated that chromosome conjunction in the sex chromosome bivalents is provided by stable SUM protein complexes. Moreover, the decrease of the diffuse nucleolar signals during normal spermatocyte maturation is consistent with the notion that focal chromosomal SUM complexes are protected against the degradation that apparently occurs with unassembled SUM proteins.

## Proteolytic cleavage of UNO is sufficient to abolish chromosome conjunction

Separase is required for homolog separation during M I in *Drosophila* males [30], and UNO includes a motif matching the consensus of separase cleavage sites (Fig 7A) [6,22]. We have previously shown that expression of UNO[nc]-EGFP, a mutant with a non-cleavable (nc) variant of the separase cleavage motif (E130A and R113A), instead of endogenous UNO (*bam> uno[nc]-EGFP* in *uno* null mutants) prevented the rapid disappearance and homolog separation during anaphase I [22]. Thus, UNO cleavage by separase was proposed to abolish homolog conjunction during wild-type M I, so that homologs can be pulled apart to opposite spindle poles. To assess whether UNO cleavage is sufficient for elimination of homolog conjunction, we generated a mutant transgene (*UASt-uno[TEV]-EGFP*), in which the separase cleavage site was replaced by three repeats of a target sequence for tobacco etch virus (TEV) protease (Fig 7A). After expression of UNO[TEV]-EGFP instead of endogenous UNO (*bam> uno[TEV]-EGFP* in *uno* null mutants), UNO[TEV]-EGFP did not disappear rapidly during anaphase I (Fig 7C), in contrast to UNO-EGFP [22]. Moreover, homolog separation was inhibited and massive chromosome bridging was observed during telophase I (Fig 7C). Thus, behavior and effects of UNO[TEV]-EGFP were identical to those previously observed with UNO[nc]-EGFP in analogous experiments [22]. Since UNO[TEV]-EGFP no longer contained the separase cleavage site, this corresponded precisely to expectations.

When a transgene driving expression of TEV protease specifically in late spermatocytes was added into the UNO[TEV]-EGFP expressing background, the resulting phenotype was very distinct. Two TEV transgenes, *exumP-TEV* and *betaTub85DP-TEV*, were generated and used. The former resulted in an onset of TEV protease effects at a slightly earlier stage compared to the latter (Fig 7B). The phenotypic effects were highly similar with both TEV transgenes and will be documented primarily for *betaTub85DP-TEV*. In *uno* null mutant spermatocytes with *bam> uno[TEV]-EGFP* and *betaTub85DP-TEV*, UNO[TEV]-EGFP disappearance occurred well

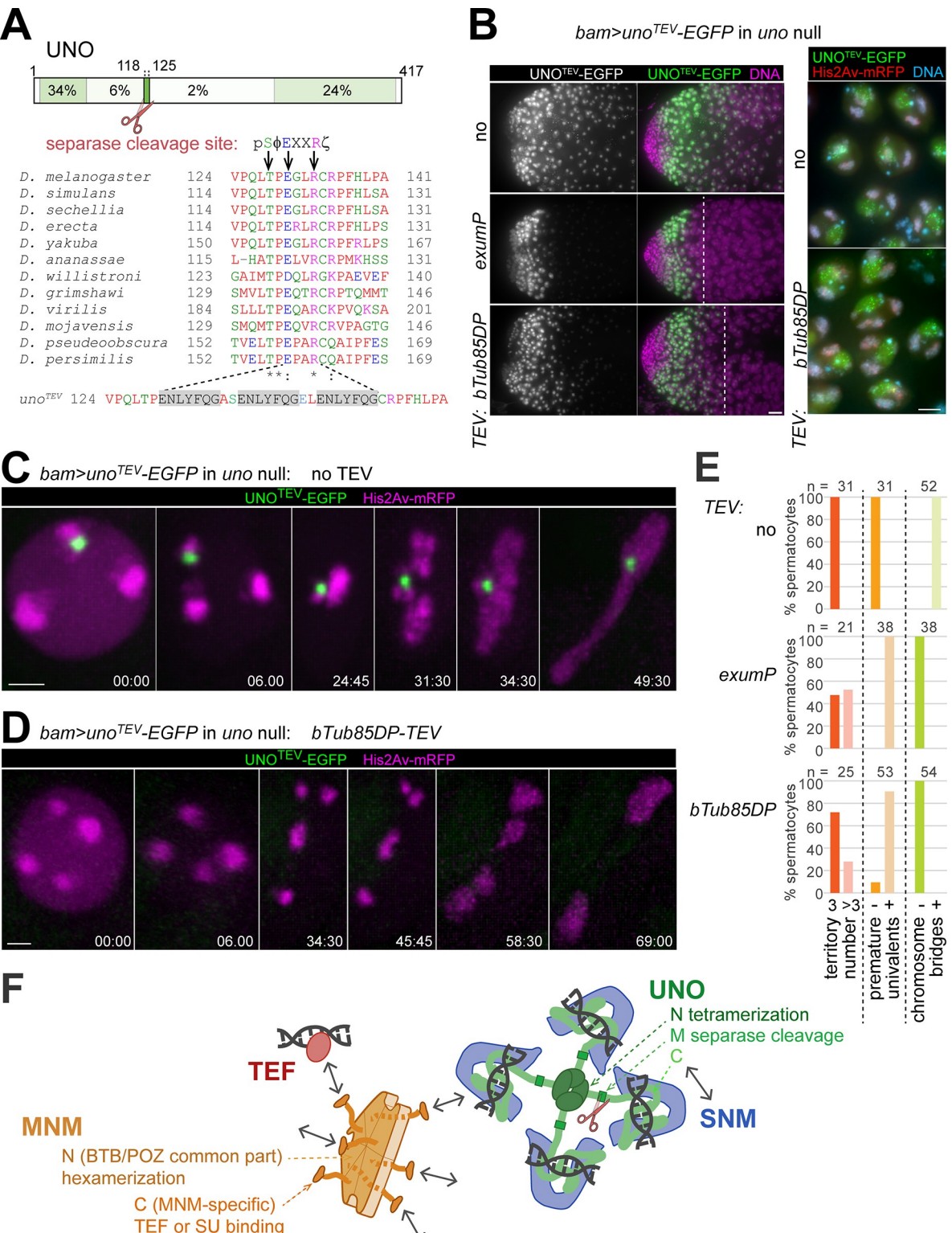

**Fig 7. UNO cleavage is sufficient for elimination of homolog conjunction.** (**A**) UNO contains a separase cleavage site preceded by a threonine (T128 in *D. melanogaster*) that is highly conserved among drosophilids [22], as illustrated with the sequence alignment. Positions essential for UNO cleavage are indicated by arrows. In the consensus sequence of separase cleavage sites [6,49], Φ indicates hydrophobic, X any amino acid and ζ hydrophilic residues. Separase cleaves C-terminally after R. The UNO^TEV-EGFP mutant contains three repeats of the recognition sequence cleaved by TEV protease (grey shading), replacing the separase cleavage site. (**B**) Squash preparations *uno* null mutant

testes with *bam>uno*^TEV*-EGFP* and either no TEV transgene, *exumP-TEV* or *betaTub85DP-TEV* (*bTub85DP-TEV*) as indicated were labeled with a DNA stain and analyzed microscopically to reveal presence and localization of UNO^TEV-EGFP during spermatocytes maturation. As indicated (dashed line) in the images with the apical testes regions on the left, a premature UNO^TEV-EGFP disappearance was induced by the TEV transgenes. This disappearance occurred after the S3 stage, which still displayed a normal chromosome territory organization and UNO^TEV-EGFP signals of normal level and localization in sub-nucleolar foci even in the presence of *betaTub85DP-TEV* (right panel). (**C**-**E**) Progression through M I in spermatocytes homozygous for the *uno*^cc1 null mutation with *His2Av-mRFP* and *bam>uno*^TEV*-EGFP* was analyzed by time-lapse imaging. The spermatocytes expressed either no TEV transgene (**C**) or *betaTub85DP-TEV* (*bTub85DP-TEV*) (**D**). Still frames of representative spermatocytes are displayed with time indicated (min:sec) with t = 0 at the onset of NEBD I. Chromosome organization at NEBD I (number of major chromosome territories), during prometaphase I (presence of univalents) and telophase I (presence of chromosome bridges) was scored. The results are displayed in the bar diagram (**E**) with numbers of analyzed spermatocytes (n) indicated. (**F**) Summary of the protein-protein and protein-DNA interactions of the AHC proteins SNM, UNO, MNM and TEF. See discussion for the implications for the mechanism of alternative homolog conjunction. Scale bars = 20 (B, left), 5 (B, right), and 3 (C,D) μm.

after the S3 stage, and S3 spermatocytes still displayed UNO^TEV-EGFP signals of normal intensity and normal localization in sub-nucleolar foci (Fig 7B). Moreover, these S3 spermatocytes also displayed normal chromosome territories (Fig 7B). Thus, *betaTub85DP-TEV* did not affect the initial establishment and maintenance of AHC. However, after the *betaTub85DP-TEV*-induced disappearance of UNO^TEV-EGFP from late spermatocytes, bivalents were prematurely separated into univalents. Cytological analyses with testis squash preparations (S7 Fig) and time-lapse imaging (Fig 7D) revealed an increase in the number of chromosome territories in S6 spermatocytes, as well as the presence of univalents during prometaphase I. After a delayed onset of anaphase I, the univalents were segregated randomly onto the two spindle poles, without formation of chromosome bridges during telophase I (Figs 7D and S7). For a quantitative confirmation of the phenotypic differences resulting from absence or presence of a TEV transgene, we scored the number of major chromosome territories around NEBD I, the presence of univalents during prometaphase I and of chromosome bridges during telophase I after time-lapse imaging (Fig 7E). Overall, these results of our experiments with UNO^TEV-EGFP suggested that proteolytic cleavage of UNO is sufficient for elimination of alternative homolog conjunction in spermatocytes.

An additional control experiment ruled out that the premature separation of bivalents after co-expression of UNO^TEV-EGFP and TEV protease in *uno* null mutants resulted from some unexpected dominant off-target effect of TEV protease unrelated to UNO^TEV-EGFP cleavage. When UNO^TEV-EGFP and TEV protease were co-expressed in spermatocytes that also produced UNO from the endogenous locus (*bam> uno*^TEV*-EGFP* and *betaTub85DP-TEV* in *uno*^-/+), bivalents were not separated prematurely (S7 Fig). Interestingly, in this *uno*^-/+ background, UNO^TEV-EGFP remained weakly detectable as a dot on the chrXY pairing region until anaphase I, while in the *uno* null mutant background, UNO^TEV-EGFP disappearance was complete already before the S6 stage (S7 Fig). As discussed below, the longer perdurance of chromosomal UNO^TEV-EGFP in the TEV expressing *uno*^+ background appears to be consistent with a mechanism for chromosome linking involving multimerized SUM protein assemblies. Also consistent with this suggestion, we observed a perdurance of weak sub-nucleolar EGFP foci until M I after expression of UNO_N- or UNO_C-EGFP in *uno*^+, but not in *uno* null mutant spermatocytes, with *UASt* transgenes and *bamP-GAL4-VP16* (S8 Fig). Expression of these UNO fragments (UNO_N- and UNO_C-EGFP) did not rescue meiotic chromosome segregation in *uno* null mutants (S8 Fig), and in the *uno*^+ background, they did not have an evident dominant-negative effect (S8 Fig), presumably as their level of expression was low according to EGFP signal intensities. Analogous expression of UNO_M-EGFP resulted in higher expression, but this fragment was present only transiently in early spermatocytes without any enrichment in the nucleolus (S8 Fig).

In an attempt to address the control of the separase-mediated UNO cleavage during M I, we generated *UASt* transgenes expressing mutant UNO-EGFP versions with alterations in a conserved potential phosphorylation site preceding the separase cleavage site (Fig 7A). Experimental evidence from fungal and mammalian organisms [12,45–49] have emphasized that α-kleisin phosphorylation promotes cleavage by separase, in particular in case of the meiotic Rec8 isoforms. The UNO orthologs of *Drosophila* species all contain a threonine residue (T128 in *D. melanogaster*) followed by a proline residue immediately upstream of the separase cleavage site (Fig 7A). Phosphorylation of a serine at an equivalent position in a fungal α-kleisin or also its phosphomimetic mutation to glutamic acid was demonstrated to stimulate separase-mediated cleavage *in vitro* [49]. To evaluate the potential role of T128 phosphorylation in UNO, we mutated the corresponding codon to encode either alanine (T128A), which cannot be phosphorylated, or the phosphomimetic aspartic acid (T128D). Expression of UNO^T128A-EGFP in *uno* null mutants (*bam>uno^T128A-EGFP* in *uno* null) resulted in homolog conjunction that resisted elimination during M I (S9 Fig). While these results supported a potential significance of T128 phosphorylation, we observed an identical phenotype unexpectedly also with the phosphomimetic UNO^T128D-EGFP (*bam>uno^T128D-EGFP* in *uno* null) (S9 Fig). While our observations demonstrate that T128 is crucial for UNO cleavage by separase during M I, further work will be required to clarify the role of phosphorylation at this position.

## Discussion

*Drosophila* male meiosis is achiasmate and therefore dependent on dedicated proteins (SNM, UNO and MNM, or SUM for abbreviation) that maintain conjunction between homologous chromosomes in replacement for the missing crossovers. Our main findings (summarized in Fig 7F) provide insight into the biochemical basis of (1) how the SUM proteins achieve this alternative homolog conjunction (AHC), and (2) how AHC is eliminated in time at the transition from metaphase to anaphase of M I to permit separation of the homologs to opposite spindle poles. In addition, our results are informative concerning the evolution of the AHC system.

We demonstrate that SNM and the C-terminal domain of UNO form a stable heterodimeric complex (SU_C). Based on sequence comparisons, AlphaFold structural predictions and XL-MS with recombinantly expressed and purified proteins, the SU_C complex is homologous to that formed by stromalin and the stromalin-binding region of α-kleisin. Stromalins and α-kleisins are components of cohesin complexes [1]. While SNM was recognized as highly similar to stromalins early on [20], the very limited similarity of UNO to α-kleisins has escaped detection until now. The important and conserved N- and C-terminal domains of α-kleisins, which mediate its binding to the SMC heterodimer in cohesin, are not present in UNO. From an α-kleisin precursor, UNO has thus retained only the stromalin-binding region and the previously identified separase cleavage site [22].

Stromalin, via positively charged surface patches, has recently been shown to promote DNA-binding of cohesin *in vitro* [43]. We find that purified SU_C also binds DNA (Fig 7F). At least one of stromalin's positively charged surface patches [43] is clearly also present in SNM and contributes to the DNA-binding of SU_C, according to our *in vitro* analysis with mutant versions of SU_C. In addition, a conspicuous, positively charged α-helix at the very C-terminus of UNO, which is absent from α-kleisins, makes a contribution to the DNA-binding of SU_C that is even more important than the basic SNM patch. Apart from DNA-binding, the interactions with the other AHC proteins were still normal in case of UNO^chm-EGFP, a mutant with acidic or neutral residues in place of the six basic residues in the C-terminal α-helix. *In vivo*, UNO^chm-EGFP displayed strongly reduced chromosome-binding and failed to

provide normal AHC during male meiosis. These results strongly argue for the physiological importance of the DNA-binding activity of SU_C. We speculate that the C-terminal α-helix of UNO might clamp down on a DNA double helix bound to the basic surface patches of SNM and thereby strongly increase the strength of DNA-binding. The binding of SU_C to DNA does not appear to be sequence specific. Clearly, in competition with the scrambled DNA sequence, we have not detected increased binding to the 240 bp repeat sequence from the rDNA intergenic spacers, which appears to mediate sex chromosome conjunction [24–26].

Beyond DNA, SU_C binds to MNM (Fig 7F). Neither SNM nor UNO interact with MNM individually, indicating that prior association of SNM and UNO is required for MNM binding. These conclusions are based on our co-immunoprecipitation experiments after transient expression in S2R+ cells. Of note, we have not accomplished SUM complex formation with purified proteins *in vitro* so far. Our attempts at expressing and purifying full length MNM were not successful. Moreover, the successfully purified C-terminal region of MNM (MNM_C), which mediates the binding to SU_C (Fig 7F) according to our co-immunoprecipitation experiments, did not bind to SU_C *in vitro*. It is conceivable, therefore, that binding of MNM to SU depends on prior post-translational processing steps. At present, our inability to generate SUM complexes *in vitro* precludes a straightforward clarification of the issue whether SU can bind simultaneously to both MNM and DNA. However, the extended contacts between the C-terminal domain of UNO and SNM over long stretches (Fig 7F) provide ample space with interface potential, thereby increasing the likelihood of simultaneous binding of MNM and DNA to SU.

MNM_C mediates binding not only to SU_C but also to TEF (Fig 7F), as revealed by the co-immunoprecipitation experiments reported here and previously [23]. The MNM-TEF interaction also remains to be re-constituted with purified proteins *in vitro*. However, in case of MNM_C, simultaneous binding of both TEF and SU_C is not feasible according to our co-immunoprecipitation experiments.

Beyond the interaction domains discussed above, our analyses demonstrated the presence of multimerization domains in both UNO and MNM. We suggest that these domains are likely of crucial importance for the molecular mechanism whereby the SUM proteins generate AHC. In case of UNO, the N-terminal domain (UNO_N), which is highly conserved in UNO homologs, self-associates (Fig 7F), forming dimers and tetramers when expressed and purified from bacteria. This UNO_N region has a predicted structure that is very distinct from that of the conserved N-terminal region of α-kleisins, indicating that the evolution of *uno* involved substitution of N-terminal in addition to deletion of C-terminal coding sequences in an ancestral α-kleisin gene. Multimerization in case of MNM is also mediated by the N-terminal region MNM_N (Fig 7F). The primary sequence of MNM_N is identical to that of the N-terminal region present in the considerable number of alternative isoforms expressed from the complex *mod(mdg4)* locus. This common part of the Mod(mdg4) protein isoforms (thus also designated as Mod(mdg4)_CP) contains a BTB/POZ domain. This protein interaction domain present in many eukaryotic proteins with diverse functions has been shown to mediate homomeric dimerization [38,39]. In addition, in case of the particular type of BTB domain that is also present in the Mod(mdg4) protein products, heteromeric and higher order multimerization has been reported based on SEC, native gel electrophoresis and crosslinking studies [50]. Our results confirm and extend these findings. Purified MNM_N/Mod(mdg4)_CP was found to form stable hexamers according to SEC-MALS. The hexamers were readily modeled by AlphaFold2 as a ring-like complex with three dimers, and negative-stain electron microscopy revealed ring-like complexes of an appropriate dimension. A recent preprint describing similar structural analyses of the same type of BTB domains (i.e. the TTK-type) derived from other

*Drosophila* proteins (Lola and CG6765) and also from Mod(mdg4) provides further confirmation of the ring-shaped hexameric structure [51].

Because of the multimerization domains (UNO_N and MNM_N/Mod(mdg4)_CP), the SUM proteins have presumably the potential to form extended protein assemblies that include many copies of the SU_C DNA-binding site (Fig 7F). Thereby, they might be empowered to effectively and stably conjoin distinct double-stranded DNA molecules (Fig 7F) and function as a chromosome glue. Accordingly, AHC would not rely on a topological ring-like embrace as proposed to be provided by cohesin in case of sister chromatid cohesion [1]. Since the tetramers and hexamers formed *in vitro* by purified UNO_N and MNM_N/Mod(mdg4)_CP are stable, SUM protein assemblies formed on bivalents in spermatocytes are expected to adopt a more solid rather than a liquid state. Indeed, our FRAP analyses confirmed that the SUM proteins in the dots associated with the sex chromosome pairing regions do not undergo dynamic exchange.

The proposed extended SUM protein assemblies with their multitude of DNA-binding sites are unlikely to conjoin exclusively homologous DNA strands. Presumably, sister DNA strands are connected as well (and perhaps even neighboring regions on the same strand). Previous characterizations of meiotic mutant phenotypes are consistent with this view. Absence of AHC function results in premature separation of bivalents into univalents in late spermatocytes and early in M I [20,22]. The SOLO and SUNN proteins, which appear to function similar to the Rec8 cohesin complexes of other eukaryotes [52–54], still assure in these univalents a functional unification of sister centromeres for organization of a single kinetochore unit, as well as well as pericentromeric sister chromatid cohesion. In *solo* and *sunn* mutants, sister centromeres and pericentromeric regions lack cohesion, but bivalents are still present until the onset of anaphase I [52,53]. As sister chromatid cohesion within the regions of chromosome arms is normally lost after territory formation already during spermatocyte maturation, the presence of bivalents in *solo* and *sunn* mutants during early M I suggests that the SUM proteins conjoin not just homologous chromatids but also sister chromatids. In support of this interpretation, *snm solo* double mutants display univalents during early M I [52]. We emphasize that a chromosomal glue that conjoins DNA strands indiscriminately, as proposed for the SUM protein assemblies (i.e., sister strands and homologous strands in *trans* and perhaps also neighboring regions in *cis*) should be perfectly adequate if it is applied at the right time during spermatocyte maturation, i.e., after disruption of non-homologous chromosomal associations by territory formation but before complete disruption of homolog associations.

Clearly, our proposal that AHC relies on extended assemblies of SUM proteins providing a high number of DNA-binding sites remains speculative and requires further investigation. For example, understanding how the formation of SUM protein assemblies is controlled and restricted to limited chromosomal regions will be crucial. The mechanism whereby SUM protein assemblies are targeted to the sex chromosome pairing rDNA loci on chromosome X and Y remains unexplained. In case of autosomal bivalents, TEF is likely involved in the initial establishment of SUM protein assemblies [23]. However, after ectopic expression in larval salivary glands, TEF as well as SUM bind to a large number of polytene chromosome bands [23]. In contrast, in mature S6 spermatocytes, the autosomal SUM protein assemblies are spatially restricted to one or two dots per chromosome arm [27]. Targeting of SUM protein assemblies to the sex chromosome pairing site and into autosomal dots might involve interactions with additional chromosomal proteins. Mod(mdg4)_T (also designated as 67.2 or 2.2), the most extensively characterized isoform expressed from the complex *mod(mdg4)* locus, interacts and co-operates with several chromatin architectural proteins (including CP190, HIPP1 and SuHw) at the gypsy insulator [37]. Moreover, Mod(mdg4)_T in combinations with chromatin architectural proteins in various combinations is generally enriched at boundaries between

topologically associated chromatin domains and also at button loci that promote the somatic pairing of homologous chromosomes [55,56]. Recently, Mod(mdg4) function has been implicated in a striking example of chromosome pairing-dependent regulation of physiological gene expression [57]. Thus, multimerization by Mod(mdg4)_CP is likely crucial for chromosomal associations other than AHC during male meiosis. Accordingly, by recruitment of the Mod(mdg4)_H isoform MNM for AHC, evolution might have co-opted a pre-adaption that achieves chromosomal associations by Mod(Mdg4)_CP multimerization.

While we found Mod(mdg4) isoforms other than MNM to be unable of binding to SU, these other isoforms clearly have the potential to form heteromeric associations with MNM according to our co-immunoprecipitation experiments. Moreover, based on yeast two-hybrid (Y2H) analyses, various other proteins with TTK-type BTB domains might also form heteromeric associations with MNM [51]. Whether such heteromeric interactions are relevant of AHC remains to be clarified. However, phenotypic analyses with various *mod(mdg4)* alleles have argued against contributions to AHC by Mod(mdg4) isoforms other than MNM [33]. Moreover, while heteromeric associations of Mod(mdg4)_T with other Mod(mdg4) isoforms can readily be detected by Y2H and co-immunoprecipitation after overexpression in S2 cells, their occurrence on chromosomes without overexpression is questionable according to chromatin-immunoprecipitation [58].

Importantly, AHC must provide conjunction between homologs in bivalents that is very robust and yet also amenable to rapid and complete elimination after biorientation of all the bivalents in the M I spindle, so that homologs can be separated to opposite poles during anaphase I. Efficient destructibility of AHC was achieved by the evolutionary co-option of the α-kleisin-derived protein UNO. Like α-kleisin, UNO includes a separase cleavage site that is highly conserved among UNO orthologs (Fig 7F) [22]. This cleavage site was shown to be required for AHC elimination and homolog separation during anaphase I [22]. Here, by exchanging the separase cleavage site in UNO with that cleaved by the bio-orthogonal TEV protease, we provide evidence that UNO cleavage is indeed sufficient to eliminate AHC. In our experiments, TEV was expressed under control of *cis*-regulatory sequences from *exu* or *betaTub85D* in mid spermatocytes. The presence of normal chromosome territories and of a normal subcellular localization of UNO$^{TEV}$-EGFP at the onset of TEV expression indicated that this TEV expression occurred after successful AHC establishment, which occurs early during spermatocyte maturation [44]. However, as a consequence of TEV expression, bivalents were prematurely converted into univalents, as clearly indicated by cytological analyses and by time lapse imaging of progression into and through M I. UNO cleavage separates the multimerization domain UNO_N from UNO_C, which mediates DNA-binding in conjunction with SNM. Therefore, we propose that UNO cleavage dissociates the chromosomal SUM protein assemblies to an extent where the number of associated DNA-binding sites is no longer sufficient for tight linkage of distinct double-stranded DNA molecules. Clearly, alternative mechanisms of AHC elimination by UNO cleavage remain conceivable, and further work will be required to clarify the mechanistic details of AHC and its elimination.

## Materials and methods

### Plasmids

For transient expression of proteins in S2R+ cells, we co-transfected pCaSper4-Act5C-GAL4 (kindly provided by Christian Klämbt, Westfälische Wilhelms-Universität, Germany) with pUASt constructs. Several of these pUASt constructs have been described previously: pUASt-snm, pUASt-mnm, pUASt-EGFP-snm, pUASt-snm-EGFP, pUASt-EGFP-mnm, pUASt-mnm-EGFP [44]; pUASt-uno-EGFP [22]; pUASt-snm-mCherry, pUASt-mnm-mCherry,

pUASt-Mod(mdg4)_CP-mCherry, pUASt-Mod(mdg4)_C-mCherry, pUASt-Mod(mdg4)_P-mCherry, pUASt-Mod(mdg4)_T-mCherry, pUASt-teflon-10xmyc, pUASt-teflon-mCherry [23]. The plasmid pUASt-nls-tetR-EGFP was kindly provided by Stefan Heidmann (Universität Bayreuth, Bayreuth, Germany).

The plasmid pUASt-uno-myc was generated by amplifying the *uno* coding region from pUASt-uno-EGFP with the primer pair OL005/ZK017 (see S1 Table for all oligonucleotide sequences). After digestion of the resulting fragment with EcoRI, it was inserted into the corresponding site of pUASt-mcs-10xmyc [23]. To generate pUASt-mnm-myc, pUASt-mcs-10xmyc was first modified. By digestion with EcoRI, followed by insertion of a linker obtained by annealing LV026/LV027, the EcoRI site was converted into a NotI site. By mutagenic plasmid amplification [59] with MS018 as primer the reading frame between the NotI site and the region coding for the myc epitopes was adjusted. Thereafter the *mnm* coding region was isolated from pUASt-mnm-EGFP as a NotI fragment and inserted into the corresponding site of the modified vector. To generate pUASt-myc-mad1_C, we enzymatically amplified the C-terminal *mad1* coding region (aa 500–730) from the cDNA clone GM14169 [60] using AF65/AF66. The fragment was digested with NotI and Acc65I, followed by ligation into the corresponding sites of pUASt, yielding pUASt-Mad1_C. For insertion of the sequences coding for 10 copies of the myc epitope, we isolated a NotI fragment from pUASt-10myc-Mps1 [61] and inserted it into the corresponding site of pUASt-Mad1_C.

For construction of pUASt-mnm_C-mCherry, we enzymatically amplified the C-terminal *mnm* region from pUASt-mnm-EGFP with ZK055/AF049. After digestion with NotI, we inserted the fragment in the corresponding site of pUASt-mcs-mCherry [23].

Additional constructs were made starting from pUASt-attB [62]. In a first step, the derivative pUASt-attB-mcs-EGFP was generated by enzymatic amplification of the EGFP coding sequence with the primer pair RAS079/RAS080, followed by digestion with KpnI/XbaI and insertion into the corresponding restriction sites of pUASt-attB. In a second step, insert fragments coding for UNO_N, UNO_M and UNO-C were amplified from pUASt-uno-EGFP with the primer pairs JW082/ZK013, ZK014/ZK015 and ZK016/JW083, respectively. After digestion with BglII/XhoI, the fragments were inserted into the corresponding sites of pUASt-attB-mcs-EGFP, yielding pUASt-attB-uno_N-EGFP, pUASt-attB-uno_M-EGFP and pUASt-attB-uno_C-EGFP. For the construction of pUASt-attb-uno_N-mCherry, we exchanged the region coding for EGFP with that encoding mCherry. The pUASt-attB derivatives for the expression of either UNO, UNO$^{TEV}$, UNO$^{T128A}$, UNO$^{T128D}$ and UNO$^{chm}$ were made starting from pUASt-attB-mCherry-uno-EGFP [23]. In a first step, the mCherry coding region was excised with EcoRI/BglII and replaced with a double-stranded DNA oligonucleotide generating by annealing of the oligos CL338/CL339, yielding pUASt-attB-uno-EGFP. For the generation of pUASt-attB-uno$^{TEV}$-EGFP, we replaced its EcoRI—AgeI fragment with the synthetic DNA fragment CL342 after digestion with the same restriction enzymes. In case of pUASt-attB-uno$^{T128A}$-EGFP and pUASt-attB-uno$^{T128D}$-EGFP, we also replaced the EcoRI—AgeI fragment of pUASt-attB-uno$^{TEV}$-EGFP with the EcoRI/AgeI-digested synthetic DNA fragments CL340 and CL341, respectively. In case of pUASt-attb-uno$^{chm}$-EGFP, the BsiWI—XhoI fragment of pUASt-attB-uno-EGFP was replaced with the BsiWI/XhoI-digested synthetic DNA fragment CL419.

To generate *exumP-TEV* transgenic flies, we modified pattB-exumP-EGFP [22] by replacing the EGFP coding region with that encoding TEV. The region coding for TEV with an N-terminal SV40 nuclear localization signal and a V5 epitope tag as well as two C-terminal SV40 nuclear localization signals was excised with EcoRI/NotI from pUASp1-TEV, a plasmid identical to that described earlier [63] except that it did not contain the S219V mutation. Thereafter,

the EcoRI—NotI fragment was used to replace the EcoRI—NotI fragment of pattB-exumP-EGFP.

The plasmid pattB-betaTub85DP-TEV contained *cis*-regulatory 5' and 3' sequences of the spermatocyte-specific *betaTub85D* gene, as also present in pattB-Nslmb-vhh4-GFP4 [44]. The region coding for TEV and the N- and C-terminal extension described above was enzymatically amplified with SCH035/NT022 and inserted between the 5' and 3' *betaTub85D* sequences after digestion with KpnI/NotI.

All constructs for expression of recombinant proteins were cloned using the InteBac system [64] for either insect cells or bacterial expression.

## Drosophila lines

Several lines with mutations or transgenes that we have used for our analyses have been described earlier: *UASt-snm-EGFP* and *UASt-mnm-EGFP* [44], *UASt-uno-EGFP* and *uno^cc1* [22], *UASt-snm-mCherry* [23], *g-His2Av-mRFP* [65], *bamP-GAL4-VP16* [66], *Sgs3-GAL4* (Bloomington *Drosophila* Stock Center (BDSC) #6870), Df(2R)Exel7094 (BDSC #7859).

Fly lines carrying the transgenes *UASt-uno^chm^-EGFP*, *UASt-uno^TEV^-EGFP*, *UASt-uno^T128A^-EGFP* and *UASt-uno^T128D^-EGFP* were generated (BestGene Inc., Chino Hills, CA, USA) by integration of the corresponding pUASt-attB constructs described above into the landing site *P{CaryP}attP2*. For production of fly lines with the transgenes *exumP-TEV*, *betaTub85DP-TEV*, *UASt-uno_N-EGFP*, *UASt-uno_M-EGFP* and *UASt-uno_C-EGFP* with the attB plasmids described above, we used the integration site *P{CaryP}attP40*.

Standard crossing and generation of recombinant chromosomes were used to produce the various strains used for experimental analyses. The genotypes of the flies analyzed are described in detail in the supporting information (S2 Table).

## Bioinformatic analyses of the predicted UNO amino acid sequence

UNO orthologs were collected with NCBI blast within the NCBI non-redundant protein database or the UniProt reference proteomes applying significant E-value thresholds below 0.001 [67]. We used HHPRED with alignments of the N- and C-terminal conserved regions of UNO for remote homology detection in the PDB and PFAM database [68]. The search with the N-terminal conserved domain was not informative, but the C-terminal region of UNO (aa 289–364) hit with a probability of 83.2 to the middle region of human Rad21 (aa 320–394) from the structure 4PJW [35]. The C-terminal hit region of UNO was aligned with the conserved middle region of Rad21, Rad21L and Rec8 orthologs using mafft (-linsi, v7.427) [69] and visualized with Jalview [70].

## Cell culture and transfection

S2R+ cells were cultured in Schneider's medium (Gibco, cat# 21720, Thermo Fisher Scientific, Waltham, MA), 10% fetal bovine serum (Gibco, cat# 10500–064) and 1% Penicillin-Streptomycin (Gibco, cat# 15140) at 25˚C. Transfections were performed using FuGENE HD (Promega, cat# E2311) in 6-well plates, T25 or T75 flasks. In case of 6-well plates, $1.2 \times 10^6$ cells were plated into one well in 2 ml complete medium. In T25 flasks, $5.2 \times 10^6$ cells were plated in 4 ml complete medium, and in T75 flasks, $15.6 \times 10^6$ cells in 8 ml complete medium. One hour after plating transfection mix was added. In case of 6-well plates, 100 µl transfection mix containing 1 µg plasmid DNA and 4 µl FuGENE HD in Schneider's medium were added. For T25 flasks, 200 µl of transfection mix was used containing 2 µg plasmid DNA and 8 µl FuGENE HD in Schneider's medium, and for T75 flasks, 400 µl of transfection mix containing 4 µg plasmid

DNA and 16 μl FuGENE HD in Schneider's medium. Cells were incubated for 2 days before analysis in co-immunoprecipitation experiments.

## Immunoprecipitation and immunoblotting

For the analysis of whole cell lysates by immunoblotting (S1 and S3 Figs), transfected cells in T25 flasks or 6-well plates were transferred on ice and carefully washed twice with cold phosphate-buffered saline (PBS) (137 mM NaCl, 2.7 mM KCl, 1.47 mM $KH_2PO_4$, 6.46 mM $Na_2HPO_4$, pH 7.4). Using a scraper, cells were harvested with a total of 150 μl resp. 60 μl of 3x Lämmli buffer (62.5 mM Tris-HCl, 10% glycerol, 5% β-mercaptoethanol, 3% SDS, 0.01% Bromophenol Blue) and the mixture was transferred to Eppendorf tubes. All samples were boiled for 8 minutes at 96˚C, aliquoted, snap frozen in liquid nitrogen and stored at -80˚C until analysis by immunoblotting.

For immunoprecipitation, the transfected S2R+ cells were detached with a cell scraper and the resulting cell suspension was centrifuged at 580 x g for 5 minutes in a 15 ml Falcon tube. The cell pellet was washed with 1 ml cold phosphate-buffered saline (PBS) (137 mM NaCl, 2.7 mM KCl, 1.47 mM KH2PO4, 6.46 mM Na2HPO4, pH 7.4) and transferred to a 1.5 ml Protein LoBind tube (Eppendorf, cat# 022431081), which were also used for all subsequent steps. Cells were sedimented at 600 x g for 5 minutes at 4˚C. All subsequent steps were performed on ice or at 4˚C with ice cold solutions. Cells were lysed in lysis buffer (LB): 20 mM Tris-HCl pH 7.5, 300 mM NaCl, 2 mM MgCl2, 0.1% Nonidet P-40 Substitute (Sigma Aldrich, cat# 74385), 5% glycerol, 0.5 mM EGTA, 1 mM DTT, 50 U/ml Benzonase Nuclease ultrapure (Sigma Aldrich, cat# E8263) and 1 tablet Roche protease inhibitor c0mplete per 10 ml lysis buffer (Mini EDTA-free, EASYpack, Roche, cat# 04693159001). The cells harvested from a T25 flask were lysed in 500 μl LB and in 1000 μl LB in case of T75 flasks by pipetting up and down twice, each time followed by a 15-minute incubation. Cell lysates were cleared by centrifugation for 15 minutes at 16100 x g. A small aliquot of the supernatant was removed for analysis by immunoblotting (input samples). The rest of the supernatant was added to pre-washed 25 μl nano-trap agarose beads (GFP-Trap agarose, ChromoTek, cat# gta-20, or RFP-Trap agarose, ChromoTek, cat# rta-20, or MYC-Trap agarose beads ChromoTek, cat# yta-20). Beads were incubated for 1 hour on a rotating wheel at 15 rpm. Thereafter, beads were washed 3 times with LB and centrifugation at 2500 x g for 2 minutes. For the final wash samples were transferred into a fresh tube. For elution, beads were resuspended in 80 μl 3x Lämmli Buffer (62.5 mM Tris-HCl, 10% glycerol, 5% β-mercaptoethanol, 3% SDS, 0.01% Bromophenol Blue) and boiled for 8 minutes at 96˚C. After rapid cooling on ice, beads were sedimented and the supernatant was distributed in three aliquots of 25 μl (IP samples). All samples were snap frozen in liquid nitrogen and stored at -80˚C until analysis by immunoblotting.

Samples were resolved by sodium dodecyl sulfate-polyacrylamide gel electrophoresis (SDS-PAGE) using a mini-gel system (BioRAD) and gels with between 7% and 12% polyacrylamide for 3 hours at 90 V. Molecular weight markers were PageRuler Plus Prestained Protein Ladder (Thermo Scientific, cat# 815-968-0747). Protein transfer to nitrocellulose membranes (Amersham Protran 0.45 μm, cat# 10600002) was achieved by tank blotting at room temperature for 1 hour with 100 V. After transfer, Ponceau S staining was performed, followed by a blocking step using 5% dry milk (w/v) in PBS with 0.02% NaN3. This solution was also used for incubation with the primary antibody for 2 hours at room temperature or overnight at 4˚C. Three washes with 5% dry milk (w/v) in PBS were done before incubation with the secondary antibody, which was applied in 5% dry milk (w/v) in PBS at room temperature for 1 hour protected from light. After 2 washes with 5% dry milk (w/v) in PBS and 2 washes with

PBS, 0.1% Tween-20, signals were detected with ECL reagents (WesternBright ECL, Advansta, cat# K-12045-D50) in an Amersham Imager 600.

The following antibodies were used for immunoblotting: rabbit polyclonal antibodies anti-GFP diluted 1:800 (ChromoTek, cat# pabg1) or 1:2000 (Torrey Pines Biolabs, cat# TP401); mouse monoclonal antibody anti-RFP (ChromoTek, cat# 6g6), 1:1000; rat monoclonal antibody anti-c-MYC (ChromoTek, cat# 9e1) at 1:1000; rabbit polyclonal antibody anti-ModC [13] (kindly provided by Rainer Dorn, Universität Halle, Halle, Germany) at 1:4000; rabbit polyclonal antibody anti-SNM C-terminal peptide (DIAHLKEYRNALRPRKTKSYPQAT) [20] (kindly provided by Bruce McKee, University of Tennessee, Knoxville, TE, USA); rabbit polyclonal antibody anti-UNO [22]; HRP-conjugated AffiniPure goat anti-rabbit IgG polyclonal antibody (Jackson ImmunoResearch, cat# 111-035-003) at 1:1000; HRP-conjugated AffiniPure goat anti-mouse IgG polyclonal antibody (Jackson ImmunoResearch, cat# 115-035-003) at 1:1000; HRP-conjugated goat anti-rat IgG antibody (Thermo Scientific, cat# 62–9520) at 1:5000.

We note that after transient expression of TetR-E and immunoblotting with anti-EGFP, we always detected two bands for unknown reasons.

## Protein expression and purification

For expression of SNM and UNO_C, baculovirus was generated as previously described [71]. Briefly, EMBacY cells were transformed with constructs for either UNO_C-Strep or MBP-SNM. Bacmids were extracted from positive transformants and used to transfect Sf9 cells. Three rounds of viral amplification were performed to produce a "V2" virus suspension. For each SNM/UNO_C prep, 2 x 400 ml flasks of Hi5 cells were infected with V2 at a 1:100 dilution (i.e. 4 ml of SNM V2 virus and 4 ml of UNO_C V2 virus), and incubated at 27°C for 72 hours. Cells were harvested, washed with PBS, and flash frozen in liquid nitrogen.

For protein purification, cells were resuspended in lysis buffer (50 mM HEPES pH 7.5, 300 mM NaCl, 5% glycerol, 0.1% Triton X-100, 1 mM $MgCl_2$ + 5 mM BME, DNAse, SERVA protease inhibitor) and lysed by sonication. Cleared lysate was loaded onto a 5 ml Strep Tactin Superflow Plus column (Qiagen), washed with lysis buffer, and eluted with elution buffer (50 mM HEPES pH 7.5, 300 mM NaCl, 5% glycerol, 1 mM $MgCl_2$ + 5 mM BME, 2.5 mM Desthiobiotin). Fractions corresponding to the SNM-UNO_C complex (SU_C) were concentrated with a 10 kDa MWCO concentrator (Pierce) and loaded onto a Superose 6 10/300 column equilibrated with storage buffer (25 mM HEPES pH 7.5, 300 mM NaCl, 5% glycerol, 1 mM $MgCl_2$ + 5 mM TCEP). Eluted fractions were concentrated and snap frozen.

N-terminally 6xHis-MBP tagged UNO_N was expressed in BL21 cells at 18°C for 16 hours grown in TB. For each purification, cells from a 1L culture were used. Harvested cells were resuspended in lysis buffer (50mM Na-HEPES, pH7.5, 300 mM NaCl, 5% Glycerol, 0.1% Triton X-100, 1 mM $MgCl_2$, 5 mM β-mercaptoethanol, DNAse, SERVA protease) using 5 ml/g pellet. Resuspended bacteria were lysed by two passages through an Emulsiflex (Avestin). Cleared lysate was passed over a 5 ml MBP Trap column (Cytiva) equilibrated in MBP-Trap A buffer (50 mM Na-HEPES, pH7.5, 300 mM NaCl, 5% Glycerol, 5 mM β-mercaptoethanol) at 2.5 ml /minute. MBP-Trap was washed with 15 CV of MBP-Trap A buffer followed by elution with 10 CV of MBP-Trap buffer B (50 mM Na-HEPES, pH7.5, 300 mM NaCl, 5% Glycerol, 1 mM Maltose 5 mM β-mercaptoethanol) collecting 5 ml fractions. Fractions corresponding to MBP-UNO_N were pooled and the NaCl concentration adjusted to 100 mM. Sample was then loaded onto a 6 ml ResourceQ column (Cytiva) pre-equilibrated with ResQ Buffer A (50 mM Na-HEPES pH7.5, 100 mM NaCl, 5% glycerol, 5 mM β-mercaptoethanol). Unbound sample was washed away with 10 CV of ResQ Buffer A. MBP-UNO_N was eluted by running a

gradient of 0–60% ResQ Buffer B (50 mM Na-HEPES pH7.5, 1 M NaCl, 5% glycerol, 5 mM β-mercaptoethanol) over 30 CVs. Fractions corresponding to MBP-UNO_N were concentrated in a Amicon 30 kDa MWCO concentrator to a volume of ~1.5 ml. Concentrated protein was loaded onto a Superdex 200 16/600 column, pre-equilibtated with SEC Buffer (50mM Na-HEPES pH7.5, 300mM NaCl, 5% Glycerol, 1mM TCEP) and run at 1 ml/minute. Fractions corresponding to the two resulting UNO_N peaks were pooled, concentrated, flash frozen, and stored at -80˚C.

N-terminally 6xHis-MBP tagged MNM_N was expressed in BL21 cells at 18˚C for 16 hours grown in TB. For each purification, cells from a 1L culture were used. Harvested cells were resuspended in lysis buffer (50mM Na-HEPES, pH7.5, 300 mM NaCl, 5% Glycerol, 0.1% Triton X-100, 1 mM MgCl$_2$, 5 mM β-mercaptoethanol, DNAse, SERVA protease) using 5 ml/g pellet. Resuspended bacteria were lysed by two passages through an Emulsiflex (Avestin). Cleared lysate was passed over a 5 ml MBP Trap column (Cytiva) equilibrated in MBP-Trap A buffer (50 mM Na-HEPES, pH7.5, 300 mM NaCl, 5% Glycerol, 5 mM β-mercaptoethanol) at 2.5 ml /minute. MBP-Trap was washed with 15 CV of MBP-Trap A buffer followed by and additional wash step with buffer containing 1 mM ATP, followed by elution with 10 CV of MBP-Trap buffer B (50 mM Na-HEPES, pH7.5, 300 mM NaCl, 5% Glycerol, 1 mM Maltose 5 mM β-mercaptoethanol) collecting 5 ml fractions. Fractions corresponding to MBP-MNM_N were pooled and the NaCl concentration adjusted to 100 mM. Sample was then loaded onto a 6 ml ResourceQ column (Cytiva) pre-equilibrated with ResQ Buffer A (50 mM Na-HEPES pH7.5, 100 mM NaCl, 5% glycerol, 5 mM β-mercaptoethanol). Unbound sample was washed away with 10 CV of ResQ Buffer A. MBP-MNM_N was eluted by running a gradient of 0–100% ResQ Buffer B (50 mM Na-HEPES pH7.5, 1 M NaCl, 5% glycerol, 5 mM β-mercaptoethanol) over 10 CVs. Fractions corresponding to MBP-MNM_N were concentrated in a Amicon 30 kDa MWCO concentrator to a volume of ~1.5 ml. The MBP moiety was removed by the addition of GST-3C protease, followed by incubation on ice for 3 hours. Concentrated protein was loaded onto a Superdex 200 16/600 column, pre-equilibtated with SEC Buffer (50mM Na-HEPES pH7.5, 300mM NaCl, 5% Glycerol, 1mM TCEP) and run at 1 ml/minute. To ensure removal of the MBP and GST-3C protease from the sample a 5 ml MBP-Trap and a 5 ml GST-Trap column were connected in-line and downstream of the Superdex 200 16/600 column Fractions corresponding to the two resulting MNM-N peaks were pooled, concentrated in an Amicon 10 kDa MWCO concentrator, flash frozen, and stored at -80˚C.

## Size exclusion chromatography

SEC experiments were run on an Akta Pure 25 system (Cytiva). Absorbances were measured at both 280 nm (blue traces) or 254 nm (red traces). Molecular weight markers (Bio-Rad) were used as a reference (grey dotted traces). Samples from SEC experiments were collected and run on SDS-PAGE gels stained with Coomassie brilliant blue.

## SEC-MALS

50 μL samples at 5–10 μM concentration were loaded onto a Superose 6 5/200 (run at 0.3 ml/min) or Superdex 75/150 (run at 0.5 ml/min) analytical size exclusion column (Cytiva) equilibrated in buffer containing 50 mM HEPES pH 7.5, 1 mM TCEP, 300 mM NaCl (for samples without nucleosomes) or 150 mM NaCl (for samples with nucleosomes) attached to an 1260 Infinity II LC System (Agilent). MALS was carried out using a Wyatt DAWN detector attached in line with the size exclusion column.

## Cross-linking mass spectrometry (XL-MS)

For XL-MS analysis proteins were diluted in 200 μL of XL-MS buffer (30 mM HEPES 6.8, 150 mM NaCl, 5% glycerol, 1 mM MgCl2, 1 mM TCEP) to the final concentration of 3 μM, mixed with 3 μL of DSBU (200 mM) and incubated for 1 hour at 25°C. The reaction was stopped by adding 20 μL of 1 M Tris-HCl pH 8.0 and incubated for another 30 min at 25°C. The cross-linked sample was precipitated by addition of 4X volumes of 100% cold acetone followed by overnight incubation at -20°C. Samples were analyzed as previously described [72]. For interaction network visualization XVis software was used and for visualization of the crosslinks on the PDB model PyXlinkViewer [73] and XMAS [74] was used. Each time a different cutoff for the cross-linking credibility was selected depending on the quality of the cross-linking data.

## Alphafold2 predictions

Predicted structures were calculated using AlphaFold Multimer (2.2.0) [40] run on GPU nodes of the Raven HPC of the Max Planck Computing and Data Facility (MPCDF), Garching. Each job was run on a single node consisting of 4 x Nvidia A100 NVlink 40 GB GPUs. Multiple predictions were generated for each run, and the best model (determined by pTM score) was then used. PAE plots were generated using a custom script (Vikram Alva, MPI Biology Tübingen).

## Mass photometry

Mass Photometry was performed in the mass photometry buffer (MP) containing 30 mM HEPES pH 7.8, 150 mM NaCl, 5% glycerol, 1 mM MgCl2, and 1 mM TCEP. Protein samples (3 μM) were pre-equilibrated for 1 hour in the MP buffer. Measurements were performed using Refeyn One (Refyn Ltd., Oxford, UK) mass photometer. Directly before the measurement, the sample was diluted 1:100 with the MP buffer. Molecular mass was determined in Analysis software provided by the manufacturer using a NativeMark (Invitrogen) based standard curve created under the identical buffer composition.

## Negative-stain electron microscopy

4 μl of MNM at 27 μg/ml were adsorbed at 25°C for 2 minutes onto glow-discharged carbon-coated grids. The grids were washed three times with water and negatively stained with three washes of 1% uranyl acetate, followed by a 5-minute incubation at 25°C. Samples were imaged with a Tecnai G2 Spirit BioTWIN microscope equipped with a LaB6 cathode operated at 120 kV. Images were recorded at low-dose conditions (19 electrons/Å2) at a corrected magnification of 82553x on a 4k × 4k CMOS camera F416 (TVIPS, Oslo, Norway).

## EMSAs

The binding reactions (10 μL volume) were carried out in EMSA buffer (25 mM HEPES pH 7.5, 0.1 μg/μL BSA, 60 mM NaCl) containing indicated fluorescently labelled DNA substrate (10 nM). The reactions were started by addition of increasing amounts of SU_C protein complexes (36.25, 72.5, 108.75, 145, 217.5, 290, 435, 580, 870, 1160 and 1740 nM) and incubated for 20 min at 30°C. After the addition of 2 μL of the gel loading buffer (60% glycerol, 10 mM Tris–HCl, pH 7.4, 60 mM EDTA, 0.15% Orange G), the reaction mixtures were resolved in 0.8% agarose gel in 1x TAE buffer (40 mM Tris, 20 mM acetic acid, 1 mM EDTA). The gels were scanned using Amersham Typhoon scanner (Cytiva) and quantified in ImageJ.

## Microscopic analyses with larval salivary glands

By standard crossing, we combined the *Sgs3-GAL4* driver with *UASt* transgenes. Wandering third instar larvae were used for dissection of salivary glands after development at 25˚C. Salivary gland preparations were made as described [75,76] with the modifications reported previously [23]. Imaging and signal quantification was also performed as described [23].

## Microscopic analyses with testis preparations

For whole-mount testis preparations, dissections from young adult males (0–1 day after eclosion) were performed in testis buffer (183 mM KCl, 47 mM NaCl, 10 mM Tris-HCl, pH 6.8). Testes were fixed in PBST containing 4% formaldehyde in 0.2 ml Eppendorf tubes for 20 minutes on a rotating wheel. Testis squash preparations were made essentially as described previously [77]. For DNA staining, testes were incubated for 10 minutes in PBS, 0.1% Triton X-100 (PBTx) containing Hoechst 33258 (1 µg/ml). After three washes with PBS, preparations were mounted under a coverslip in a drop of mounting medium. Microscopic quantification of the DNA content of nuclei of early round spermatids was done as described [44].

Preparations of ovaries and testes were analyzed with a wide-field fluorescence microscope (Zeiss Axio Observer HS) using 40×/1.3, 63×/1.4 and 100×/1.4 oil immersion objectives. Maximum intensity projections of image stacks are presented.

Time-lapse imaging of progression through meiosis was performed as described [78]. Testes were dissected from pupal or young adult males in Schneider's *Drosophila* Medium (Invitrogen, #21720) supplemented with 10% fetal bovine serum (Invitrogen) and 1% penicillin/streptomycin (Invitrogen, #15140). The dissected pupal testes were transferred into 45 µl of medium in a 35 mm glass bottom dish (MatTek Corporation, #P35G-1.5-14-C) and opened with fine tungsten needles to release the cysts. In case of adult testes, 150 µl of medium were used. To reduce sample movements, 15 µl of 1% w/v methylcellulose (Sigma, #M0387) was added to pupal testes preparations and 50 µl to adult testes preparations. A wet filter paper was placed inside along the dish wall before sealing the lid with parafilm.

Imaging was performed at 25˚C in a room with temperature control using a spinning disk confocal microscope (VisiScope with a Yokogawa CSU-X1 unit combined with an Olympus IX83 inverted stand and a Photometrics evolve EM 512 EMCCD camera, equipped for red/green dual channel fluorescence observation; Visitron systems, Puchheim, Germany). A 60×/1.42 oil immersion objective was used. We acquired z-stacks with 30–40 focal planes spaced by 500 nm at 45-second intervals.

Maximum intensity projections were generated using ImageJ or ZEN software for wide-field images and IMARIS (Bitplane) for spinning disk confocal images. Figures display maximum intensity projections unless stated otherwise. Export of projections from IMARIS as movies or still frames after live imaging was made with interpolated image display. Moreover, display parameters for the His2Av-mRFP were adjusted manually over time to reveal chromosomes clearly throughout the movies, thereby correcting photobleaching and partially also the changes in the extent of chromosome condensation during M I. Graphs were generated with Microsoft Excel or GraphPad Prism. P values were calculated using a two-tailed student t-test (* = $p < 0.05$; ** = $p < 0.01$; *** = $p < 0.001$). Adobe Photoshop and Adobe Illustrator were used for production of figures.

## FRAP analyses

Testes were isolated from early pupae and cysts were released for imaging in dishes with a glass coverslip at the bottom as described [78]. Before bleaching, we acquired five z-stacks with 40 optical sections spaced by 500 nm at one-minute intervals using a FV1000 Olympus laser

scanning confocal microscope with a PLAPON 60XO/1.42 Objective with a zoom factor of 5.4. Thereafter, we photobleached a part of the nucleolus in one of the spermatocytes by using an circular region of interest (diameter = 30 pixels) and 100 iterations of tornado scanning with maximal 488 nm laser intensity within one z section. After photobleaching the EGFP signals in part of a nucleolus, imaging of z-stacks was continued, initially as before photobleaching. However, after five z-stacks, the time interval between z-stack acquisition was increased from one to 15 minutes. In case of long-term FRAP analyses with *bam>mnm-EGFP* spermatocytes, we acquired only three z-stacks at one-minute intervals immediately after photobleaching, followed by acquisition of three additional z-stacks at one-minute intervals four hours later.

For the quantitative analysis of EGFP signal recovery after photobleaching over time (S6 Fig), we used spot detection by IMARIS software to identify spheres containing the bleached nucleolus or unbleached nucleoli in neighboring spermatocytes of the imaged cyst. After creating a first set of spheres with a diameter of five μm, a second set of spheres with a diameter of seven μm was generated for background correction of signal intensities in the smaller spheres. For further processing with Microsoft Excel, the signal intensities detected in the spheres were exported from IMARIS. The difference in signal intensities observed in the large and small sphere, respectively, was used for estimation of background signal intensity, which was subtracted from the total intensity value within the small sphere. Moreover, fluorescence intensities were normalized to the average detected during the five z-stacks acquired before the bleaching of a nucleolus.

For the quantification of the overall recovery of EGFP signals 90 minutes after photobleaching (Fig 6A and 6B), we generated intensity sum projections using Image J. Representative projection images are shown in Fig 6A. A circular ROI with a diameter of 30 pixels, as used before for photobleaching, was placed over the bleached region. Moreover, a bean-shaped ROI covering the unbleached part of the targeted nucleolus was selected manually. Average pixel intensities in these ROIs were quantified for the five pre-bleaching time points, as well as for the first and last post-bleaching time points. Pixel intensities were normalized to the average of the five pre-bleaching time points. The intensity difference between the first and last post-bleaching time points observed in the non-bleached and bleached parts of the nucleolus were compared to estimate the extent of FRAP corrected for photobleaching during image acquisition after pulse-bleaching of a part of a nucleolus. To express the extent of FRAP after 90 minutes in percent of the total signal intensity loss induced by the pulse-bleaching within the bleached part of the nucleolus (Fig 6B), this total signal intensity loss was calculated as the difference between the intensities at the first post-bleaching time point in the non-bleached and bleached region, respectively, of the targeted nucleolus.

To compare the intensity of EGFP signals that were either diffusely distributed throughout the nucleolus or within sub-nucleolar foci, we analyzed intensity sum projections of the pre-bleaching z-stacks with Image J. The projection images were segmented by using the threshold tool of Image J and selecting first the top 2% intensity pixels, which represented the sub-nucleolar foci well. Thus, the resulting intensity values were used as a measure of signal intensity in sub-nucleolar foci. Thereafter, a second segmentation was applied for selection of the top 10% intensity pixels, which covered the complete nucleoli. To estimate the diffuse nucleolar signals, we subtracted the intensities detected within the sub-nucleolar foci from those of the complete nucleoli.

## Supporting information

**S1 Fig. Characterization of endogenous expression of SNM, UNO and MNM in S2R+ cells by immunoblotting.** (A-C) Total extracts of S2R+ cells, either untransfected or transfected for

transient expression of EGFP-SNM (E-SNM), SNM-EGFP (SNM-E), UNO-EGFP (UNO-E), EGFP-MNM or MNM-EGFP (MNM-E) were analyzed by immunoblotting with the indicated antibodies. Prestained marker proteins with the indicated molecular weights (kDa) are displayed on the left. Relative amounts of extract loaded are indicated on top. Ponceau S staining of the membrane after protein transfer for control of loading is presented below the immunoblots. (**A**) A band indicating expression of endogenous SNM is not detected by anti-SNM, in contrast to transiently expressed SNM-EGFP, confirming antibody functionality. (**B**) A band indicating expression of endogenous UNO is not detected by anti-UNO, in contrast to transiently expressed UNO-EGFP, confirming antibody functionality. (**C**) An antibody (anti-M_CP) against the common part present in MNM and the other isoforms expressed from the *mod(mdg4)* locus detects bands in untransfected S2R+ cells that are much weaker than those reflecting transiently expressed MNM-EGFP. The anti-M_CP immunoblot is shown after short (top) and long exposure (bottom) to reveal these weak bands.
(PDF)

**S2 Fig. Sequence similarity of UNO and α-kleisins.** An alignment of the predicted amino acid sequences of the indicated proteins reveals similarities between a region close to the C-terminus of UNO and an internal α-kleisin region, which is known to bind to stromalin/SA/STAG proteins. The family of α-kleisins includes members that function in meiosis (Rec8 proteins) beyond those functioning also during mitosis (Rad21 and Rad21L proteins). While the genome of *D. melanogaster* does not contain a canonical *rec8* ortholog, it includes *c(2)M*, a more distant meiotic α-kleisin [79]. Accession numbers from the UniProt database (*Dmel_uno*, *Dere_uno*, *Dpse_uno*, *Mdom_uno*) or the NCBI database (all others) are provided next to the gene names with species abbreviated: *Drosophila melanogaster* (*Dmel*), *Drosophila erecta* (*Dere*), *Drosophila pseudoobscura* (*Dpse*), *Musca domestica* (*Mdom*), *Ceratitis capitata* (*Ccap*), *Bactrocera dorsalis* (*Bdor*), *Homo sapiens* (*Hsap*), *Mus musculus* (*Mmus*). Residues (triangles) of the human α-kleisin Rad21 that directly interact with human SA1 [36], as well as the α-helices (magenta rods) in Rad21 are indicated below the alignment.
(PDF)

**S3 Fig. Comparison of protein levels after individual or combined expression of SNM and UNO.** Co-expression of SNM with UNO or with the UNO_C fragment results in higher expression levels in comparison to the individual expression of these proteins. Levels of tagged proteins resulting after transient expression of UNO (**A**) or UNO fragments (**B**) together with the indicated proteins in S2R+ cells were analyzed by immunoblotting. SNM was tagged with mCherry (SNM-C). UNO (**A**) and the fragments UNO_N, UNO_M and UNO_C (**B**) were tagged with EGFP (UNO-E, UNO_N-E, UNO_M-E and UNO_C-E). A plasmid for expression of a C-terminal fragment of Mad1 tagged with a myc-epitope (myc-Mad1_C) was co-transfected for comparison of transfection efficiencies. Total cell extracts were resolved and analyzed with anti-mCherry (anti-C), anti-EGFP (anti-E), and anti-myc (anti-myc). Relative amounts of extract loaded are indicated on top. Ponceau S staining of the membrane after protein transfer for control of loading is presented below the immunoblots. The positions of prestained marker proteins with the indicated molecular weights (kDa) and of the bands representing the indicated proteins of interest are displayed on the right.
(PDF)

**S4 Fig. Purification and structural analysis of UNO_N.** (**A**) Final purification step of UNO_N. The SEC profile revealed two peaks (Peak 1 and Peak 2), both containing only UNO_N according to SDS-PAGE and Coomassie staining. (**B**) SEC-MALS of the UNO_N complexes in Peak 1 and Peak 2 resulted in the indicated molecular masses. (**C**) Topology map

of a UNO_N monomer. (**D**) AF2 model of the UNO_N dimer with predicted alignment error shown below. (**E**) AF2 model of the UNO_N tetramer with predicted alignment error shown below.
(PDF)

**S5 Fig. Purification and structural analysis of N-terminal part of MNM.** (**A**) Final purification step of Mod(mdg4)_CP, the N-terminal part of MNM, which is also present in other Mod (mdg4) protein isoforms. Analysis by SDS-PAGE and Coomassie staining indicates that the peak obtained by SEC contains pure Mod(mdg4)_CP in a multimerized form. (**B**) Molecular mass determination of purified Mod(mdg4)_CP by SEC-MALS resulted in a value of 80.92 kDa, consistent with a hexamer formation. (**C**) AF2 model of Mod(mdg4)_CP hexamer with predicted alignment error shown below. (**D**) Ribbon diagram of one of the three Mod(mdg4) _CP dimers that form the hexamer according to the AF2 model. (**E**) Surface electrostatics on the Mod(mdg4)_CP hexamer showing a lack of obvious DNA-binding regions. (**F**) NS-EM image of the Mod(mdg4)_CP hexameric rings.
(PDF)

**S6 Fig. FRAP analysis with spermatocytes expressing SNM-, UNO- or MNM-EGFP.** FRAP analyses were completed with S5 cysts released from pupal testes of males expressing the indicated *UASt* transgenes driven by *bam-GAL4-VP16*. A subregion of the nucleolus was bleached in one of the spermatocytes of the cyst, while neighboring spermatocytes were used as controls, as illustrated with the still frames acquired during an experiment with SNM-EGFP just before and after the bleaching. The timeline on top illustrates the image acquisition sequence. EGFP signal intensities in the nucleoli were quantified and are plotted after normalization to the average of the intensities observed before bleaching.
(PDF)

**S7 Fig. Localization and effects of UNO$^{TEV}$-EGFP in different genotypes.** (**A**) Phenotypic characterization with squash preparations of testes from *uno* null mutants with *bam>uno$^{TEV}$-EGFP* and either no TEV transgene or *betaTub85DP-TEV* (*bTub85DP-TEV*) provided further confirmation of the findings revealed by time-lapse imaging. In the absence of a TEV transgene UNO$^{TEV}$-EGFP was readily detectable at the S6 stage with a sub-cellular localization identical to that of wild-type UNO-EGFP [22]. The S6 spermatocytes also displayed a normal number of major chromosome territories. However, in the presence of *bTub85DP-TEV*, UNO$^{TEV}$-EGFP was not detectable in S6 spermatocytes, which displayed an increased number of chromosome territories. Each micrograph is shown twice, on the left with enhanced EGFP signal intensities, which reveal weak autosomal UNO$^{TEV}$-EGFP dots in the absence of a TEV transgene and still no signals in the presence of *bTub85DP-TEV*. Still frames from cysts during telophase I analyzed by time-lapse imaging (bottom), illustrate the presence of massive chromosome bridges with persisting non-degradable UNO$^{TEV}$-EGFP, but only in the absence of a TEV transgene. (**B**) Still frames after time-lapse imaging of *His2A-mRFP* expressing *uno* null mutants with *bam>uno$^{TEV}$-EGFP* and *exumP-TEV* during M I reveal premature bivalent separation and absence of chromosome bridges during telophase I. Time (min:sec) with t = 0 at the onset of NEBD I is indicated. (**C**) Comparison of UNO$^{TEV}$-EGFP signal intensities at the onset of NEBD I in indicated genotypes after time-lapse imaging. While UNO$^{TEV}$-EGFP is completely eliminated by *bTub85DP-TEV* in *uno$^{cc1}$* homozygous null mutant spermatocytes (*uno$^{-/-}$*), residual UNO$^{TEV}$-EGFP is detectable in *uno$^{cc1}$* heterozygous spermatocytes (*uno$^{-/+}$*) despite the presence of *bTub85DP-TEV*. (**D**) Disappearance of residual UNO$^{TEV}$-EGFP (arrows) during anaphase I and absence of chromosome bridges during telophase I in *uno$^{-/+}$* spermatocytes with *bam>uno$^{TEV}$-EGFP* and *bTub85DP-TEV*, as revealed by time-lapse

imaging of spermatocytes expressing *His2Av-mRFP*. Time (min:sec) with t = 0 at the onset of NEBD I is indicated. Scale bars = 10 (A) and 3 (B-D) μm.
(PDF)

**S8 Fig. Expression and localization of the UNO fragments UNO_N-, UNO_M- and UNO_-C-EGFP in spermatocytes.** (**A**) The driver *bamP-GAL4-VP16* was used for expression of the indicated *UASt* transgenes in an *uno*[+] background. Testis squash preparations for comparison of expression level and pattern were labeled with a DNA stain. Apical testes regions are displayed. The images presenting merged DNA and EGFP channels (top row) are shown with identical settings during acquisition and display, indicating that expression levels were maximal in case of UNO_M-EGFP, weaker for UNO_N-EGFP and considerably weaker for UNO_C-EGFP. In case of the images displaying only the EGFP signals in grey values (bottom row), display settings were enhanced in case of UNO_C-EGFP to reveal its weak expression. (**B**) After peak expression of UNO_N-, UNO_M- and UNO_C-EGFP in early spermatocytes (see panel A), maintenance at low levels in late spermatocytes was detectable in case of UNO_-N-EGFP and UNO_C-EGFP, as documented by high resolution of images from testes squash preparations. While at the S5 stage, UNO_N-EGFP and UNO_C-EGFP were in sub-nucleolar foci in spermatocytes heterozygous for the *uno*[cc1] null allele (*uno*[-/+]) (comparable to full-length UNO-EGFP, see Fig 5F), they displayed an abnormal diffuse nucleolar localization in *uno*[cc1] homozygous spermatocytes (*uno*[-/-]). At the S6 stage, UNO_N-EGFP and UNO_C-EGFP were in weak dots on the sex chromosome bivalent in *uno*[-/+] spermatocytes and not detectable in *uno*[-/-] spermatocytes. Scale bars = 20 (A) and 5 (B) μm.
(PDF)

**S9 Fig. UNO[T128A]-EGFP and UNO[T128D]-EGFP promote homolog conjunction that cannot be eliminated during anaphase I.** (**A,B**) Time-lapse imaging of progression through M I with His2Av-mRFP expressing spermatocytes was performed for the characterization of phenotypic consequences of the T128A (**A**) and T128 (**B**) mutations that alter a conserved potential phosphorylation site immediately upstream of the separase cleavage site in UNO (see Fig 7A). The mutants were expressed in *uno* null mutant spermatocytes with *UASt* transgenes and the driver *bamP-GAL4-VP16*. Time (min:sec) with t = 0 at the onset of NEBD I is indicated. Scale bars = 2 μm.
(PDF)

**S1 Movie. Progression through M I in *uno* null mutant spermatocyte with *bam>uno*[chm]-*EGFP* and *His2Av-mRFP*.** Time-lapse analysis was used for analysis of progression through M I. Maximum intensity projections of z-stacks acquired at 45 sec intervals of the spermatocyte, which is also displayed in Fig 5H, are shown.
(MP4)

**S2 Movie. Progression through M I in *uno* null mutant spermatocyte with *bam>uno*[TEV]-*EGFP* and *His2Av-mRFP*.** Time-lapse analysis was used for analysis of progression through M I. Maximum intensity projections of z-stacks acquired at 45 sec intervals of the spermatocyte, which is also displayed in Fig 7C, are shown. The images sequence is presented three times. During the first period, display settings do not saturate the strong UNO[TEV]-EGFP dot on the chrXY bivalent. During the second period, green signal intensities are enhanced to reveal the weak UNO[TEV]-EGFP dots on autosomal bivalents. During the final repetition, only the enhanced green signals are displayed as grey values.
(MP4)

**S3 Movie. Progression through M I in *uno* null mutant spermatocyte with *bam>uno^TEV*-*EGFP*, *betaTub85DP-TEV* and *His2Av-mRFP*.** Time-lapse analysis was used for analysis of progression through M I. Maximum intensity projections of z-stacks acquired at 45 sec intervals of the spermatocyte, which is also displayed in Fig 7D, are shown. For comparison with S2 Movie, the images sequence is also presented three times with distinct display settings, as described for S2 Movie, even though UNO-TEV-EGFP is not detectable during M I in this genotype.
(MP4)

**S4 Movie. Progression through M I in *uno^-/+* mutant spermatocyte with *bam>uno^TEV*-*EGFP*, *betaTub85DP-TEV* and *His2Av-mRFP*.** Time-lapse analysis was used for analysis of progression through M I. Maximum intensity projections of z-stacks acquired at 45 sec intervals of the spermatocyte, which is also displayed in S7D Fig, are shown. For comparison with S2 Movie, the images sequence is also presented three times with distinct display settings, as described for S2 Movie. A weak UNO^TEV-EGFP dot on the chrXY bivalent is barely detectable with the display settings during the first period, but it is readily apparent with the enhanced settings during the second and third period. UNO^TEV-EGFP dot on autosomal bivalents cannot be detected even with the enhanced display settings.
(MP4)

**S1 Table. Synthetic DNA fragments.**
(XLSX)

**S2 Table. Description of the analyzed genotypes.**
(XLSX)

**S3 Table. Source data.**
(XLSX)

## Acknowledgments

We thank Rainer Dorn and Bruce McKee for providing antibodies, Katharina Hipp (MPI for Biology, Tübingen, Germany) for help with NS-EM experiments, Franziska Müller and Petra Janning (MPI of Molecular Physiology, Dortmund, Germany) for the XL-MS data sets, Veronika Altmannova (Weir Lab) for help with EMSA experiments, as well as Sina Moser and Hiro Yamada for technical help.

## Author Contributions

**Conceptualization:** Zeynep Kabakci, John R. Weir, Christian F. Lehner.

**Data curation:** Zeynep Kabakci, John R. Weir, Christian F. Lehner.

**Formal analysis:** Zeynep Kabakci, Alexander Schleiffer, John R. Weir, Christian F. Lehner.

**Funding acquisition:** John R. Weir, Christian F. Lehner.

**Investigation:** Zeynep Kabakci, Heidi E. Reichle, Bianca Lemke, Dorota Rousova, Samir Gupta, Joe Weber, Alexander Schleiffer, John R. Weir, Christian F. Lehner.

**Project administration:** John R. Weir, Christian F. Lehner.

**Supervision:** John R. Weir, Christian F. Lehner.

**Validation:** Zeynep Kabakci, John R. Weir, Christian F. Lehner.

**Visualization:** Zeynep Kabakci, Samir Gupta, Alexander Schleiffer, John R. Weir, Christian F. Lehner.

**Writing – original draft:** Zeynep Kabakci, John R. Weir, Christian F. Lehner.

**Writing – review & editing:** Zeynep Kabakci, John R. Weir, Christian F. Lehner.

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
