## [Decision Letter · Decision Letter 0]

18 Nov 2022

Dear Christian,

Thank you very much for submitting your Research Article entitled 'Homologous chromosomes are stably conjoined for *Drosophila*male meiosis I by SUM, a multimerized protein assembly with modules for DNA-binding and for separase-mediated dissociation co-opted from cohesin' to PLOS Genetics.

The manuscript was fully evaluated at the editorial level and by two independent peer reviewers. Reviewer 1 strongly urged ACCEPT, while reviewer 2 indicated several points that, if clarified, would likely improve an already very nice paper. Only one of these point suggests a need for additional experiments.  I urge you to carefully consider these matters and revise the paper as you see best fit, and to the degree that is possible. Once you are done, please return the paper as directed below. with a mercifully brief Response to  the Reviewers' Comments .  I feel comfortable making the final decision at that point; the paper will not likely go out for re-review.

It has been a honor to work with you during this process.

We therefore ask you to modify the manuscript according to the review recommendations. Your revisions should address the specific points made by each reviewer.

Yours sincerely,

R. Scott Hawley

Academic Editor

PLOS Genetics

Gregory P. Copenhaver

Editor-in-Chief

PLOS Genetics

Reviewer's Responses to Questions

**Comments to the Authors:**

Reviewer #1: review uploaded as an attachement

Reviewer #2: Homolog interactions in Drosophila male meiosis are stabilized and maintained by a group of proteins that includes SNM, UNO and MNM. This paper characterizes the interactions between these proteins. SNM is found to bind to the C-terminal region of UNO. The most interesting and significant aspects of this paper is the finding that UNO has some features of the alpha-Kleisin subunit of cohesin. This is consistent with sequence data showing SNM is a paralog of Stromalin, as well as modeling of the SNM-UNO structure, and cross-linking data. This complex also binds DNA. UNO also has a conserved separase cleavage site but not the sequences that interact with SMC1/3. This is consistent with previous work showing the SMC1/3 are not require for homolog interactions in male meiosis.

There is very little to criticize in this paper. Much of the data is describing the interactions between these proteins and some of the interpretations are limited. The DNA binding activity does not show any specificity. The data does not lead to a model of how these proteins hold homologs together, besides an unsatisfying suggestion that multimerization allows the structure to hold homologs together, like a giant glob of glue. This hypothesis arise from nice data showing that MNM self associates and forms multimers. Neither of these are serious concerns, but do show the limits of the mostly descriptive nature of the results.

1) Figures 1 and 2 are pretty dense, but I also don’t have a good suggestion on how to improve them.

2) Line 90-91: it may be worth being more specific that centromeric cohesion depends on protecting meiotic kleisins like Rec8 with proteins like MEI-S332 and DMT that recruit PP2A.

3) Line 290: the computational prediction looks like a good working model; “correct” is an overstatement.

4) Pg 15: the DNA content assay on this page for chromosome segregation is crude. There are much better and accurate cytological assays, including some previously published by this PI.

5) The results in Figure 7 are not very surprising. If there was a place to shorten the paper, it would be here. The reason is that it was already known that UNO was a target of separase mutating the cleavage site resulted in defects in homolog separation. Replacing the separase site with a TEV site is essentially the same experiment. Then adding TEV protease only shows what happens when UNO is degraded. It is not a biologically important results because degrading UNO by any number of methods would result in the same phenotype. For example, targeting SNM with TEV might have the same effect.

**Have all data underlying the figures and results presented in the manuscript been provided?**

Reviewer #1: Yes

Reviewer #2: Yes

PLOS authors have the option to publish the peer review history of their article (what does this mean?). If published, this will include your full peer review and any attached files.

Reviewer #1: No

Reviewer #2: No

---

## [Editor Report · Decision Letter 1]

28 Nov 2022

Dear Dr Lehner,

We are pleased to inform you that your manuscript entitled "Homologous chromosomes are stably conjoined for *Drosophila* male meiosis I by SUM, a multimerized protein assembly with modules for DNA-binding and for separase-mediated dissociation co-opted from cohesin" has been editorially accepted for publication in PLOS Genetics. Congratulations!

Yours sincerely,

R. Scott Hawley

Academic Editor

PLOS Genetics

Gregory P. Copenhaver

Editor-in-Chief

PLOS Genetics

Comments from the reviewers (if applicable):

**Data Deposition**

http://datadryad.org/submit?journalID=pgenetics&manu=PGENETICS-D-22-01170R1

**Press Queries**

---

## [Editor Report · Acceptance letter]

5 Dec 2022

PGENETICS-D-22-01170R1 

Homologous chromosomes are stably conjoined for *Drosophila* male meiosis I by SUM, a multimerized protein assembly with modules for DNA-binding and for separase-mediated dissociation co-opted from cohesin 

Dear Dr Lehner, 

We are pleased to inform you that your manuscript entitled "Homologous chromosomes are stably conjoined for *Drosophila* male meiosis I by SUM, a multimerized protein assembly with modules for DNA-binding and for separase-mediated dissociation co-opted from cohesin" has been formally accepted for publication in PLOS Genetics! Your manuscript is now with our production department and you will be notified of the publication date in due course.

With kind regards,

Anita Estes

PLOS Genetics

On behalf of:
